# Selective EMC subunits act as molecular tethers of intracellular organelles exploited during viral entry

Parikshit Bagchi [1✉], Mauricio Torres[2], Ling Qi[2] & Billy Tsai[1✉]

Although viruses must navigate the complex host endomembrane system to infect cells, the strategies used to achieve this is unclear. During entry, polyomavirus SV40 is sorted from the late endosome (LE) to the endoplasmic reticulum (ER) to cause infection, yet how this is accomplished remains enigmatic. Here we find that EMC4 and EMC7, two ER membrane protein complex (EMC) subunits, support SV40 infection by promoting LE-to-ER targeting of the virus. They do this by engaging LE-associated Rab7, presumably to stabilize contact between the LE and ER. These EMC subunits also bind to the ER-resident fusion machinery component syntaxin18, which is required for SV40-arrival to the ER. Our data suggest that EMC4 and EMC7 act as molecular tethers, inter-connecting two intracellular compartments to enable efficient transport of a virus between these compartments. As LE-to-ER transport of cellular cargos is unclear, our results have broad implications for illuminating inter-organelle cargo transport.

[1] Department of Cell and Developmental Biology, University of Michigan Medical School, 109 Zina Pitcher Place, BSRB 3043, Ann Arbor, MI 48109, USA.
[2] Department of Molecular and Integrative Physiology, University of Michigan Medical School, Brehm Tower Rm 5325, 1000 Wall St., Ann Arbor, MI 48105, USA. ✉email: pabagchi@umich.edu; btsai@umich.edu

During entry, a virus charts a course through the convoluted endomembrane system of the host cell to reach the cytosol and, in many cases, the nucleus leading to productive infection[1,2]. In parallel, it must avoid a degradative fate that results in non-productive infection. How this challenging task is accomplished remains enigmatic. This study reveals the molecular basis of a key intracellular membrane transport step during productive entry of polyomavirus (PyV). PyVs cause devastating human diseases, especially in immunocompromised patients. Prominent human PyVs include the BK polyomavirus (BK PyV) that induces hemorrhagic cystitis and nephropathy, JC polyomavirus (JC PyV) that causes progressive multifocal leukoencephalopathy, and the Merkel cell PyV that triggers Merkel cell carcinoma[3,4]. The simian SV40 is the archetype PyV, possessing structural and genetic similarities to human PyVs, as well as sharing a similar infection pathway as its human counterparts. Not surprisingly, studies on SV40 entry have illuminated the cellular basis of human PyV infection.

Structurally, SV40 consists of 72 pentamers of the capsid protein VP1 that encases its DNA genome, with each pentamer harboring an internal protein VP2 or VP3[5–7]. When assembled, the viral particle displays a diameter of 45 nm. To infect cells, SV40 binds to ganglioside GM1 receptor on the plasma membrane (PM)[8–11], is endocytosed, and targets to the late endosome (LE)[12]. The virus then sorts to the endoplasmic reticulum (ER) by a poorly-defined mechanism[13–16]. ER-arrival of PyV is a critical entry step because from this compartment, the virus can penetrate the ER membrane to access the cytosol[17]. Upon reaching the cytosol, PyV mobilizes further to the nucleus[18,19] where transcription and replication of the viral genome lead to lytic infection or cellular transformation[20]. How SV40 targets from the LE to the ER, a decisive infection step, remains mysterious.

Combining cell-based, biochemical, and microscopy approaches, in the context of loss-of-function conditions, we demonstrate that EMC4 and EMC7, two subunits of the ER membrane protein complex (EMC), support SV40 infection by promoting LE-to-ER targeting of the virus. EMC4 and EMC7 accomplish this by engaging the cytosolic Rab7 GTPase bound to the LE membrane, thereby supporting membrane contact between the LE and ER. EMC4 and EMC7 also interact with the ER-resident fusion machinery component syntaxin18, which is essential for ER-arrival of SV40. These findings suggest that EMC4 and EMC7 act as molecular tethers, connecting two different intracellular organelles to enable efficient transport of a viral particle between these compartments required for infection. More broadly, as the ER is known to establish membrane contact with the LE that subserves specific cellular activities[21], our identification of two host components that likely support this membrane contact site should have general implications for understanding inter-organelle communication.

## Results

**EMC4, EMC6, and EMC7 promote SV40 infection**. We previously used an acute siRNA-mediated, knockdown approach coupled to a functional rescue strategy and identified EMC1 of the ten-subunit EMC complex as a host component that promotes ER-to-cytosol membrane transport of SV40[22]. In this same study, our knockdown approach also suggested that three additional EMC subunits—EMC4, EMC6, and EMC7—play significant roles during SV40 infection. To unambiguously establish a role of these EMC subunits in SV40 infection, we asked if expressing an siRNA-resistant construct under the knockdown condition can rescue the block in SV40 infection caused by depletion of either EMC4, EMC6, or EMC7. Accordingly simian CV-1 cells, used classically to study SV40 entry, were transfected with either a control (scrambled) or EMC4 siRNA. Cells were subsequently transfected with the control plasmid GFP-FLAG or a EMC4-FLAG construct; the EMC4-FLAG construct is resistant to the EMC4 siRNA because this siRNA is designed against the 5' UTR region of EMC4. Only cells expressing FLAG were scored for expression of large T antigen, a virally-encoded protein expressed only upon successful entry of the virus into the host nucleus. Using this approach, we found that whereas knockdown of EMC4 blocked SV40 infection, expressing EMC4-FLAG under the EMC4 knockdown condition fully restored virus infection (Fig. 1a). Likewise, expression of siRNA-resistant EMC6 (EMC6∗-FLAG) in EMC6-depleted cells fully restored the block in virus infection due to depletion of EMC6 (Fig. 1b), while expressing siRNA-resistant EMC7 (EMC7∗-FLAG) in EMC7-depleted cells completely rescued the decrease in SV40 infection caused by knockdown of EMC7 (Fig. 1c). These results unequivocally establish a role of EMC4, EMC6, and EMC7 in SV40 infection.

As knockdown of a single subunit within a multisubunit complex can decrease the stability of the other subunits, we asked if depletion of either EMC4, EMC6, or EMC7 affects the level the other EMC subunits. In CV-1 (and HEK 293 T) cells, the EMC4, EMC6, and EMC7 siRNAs decreased expression of the targeted EMC subunit without significantly affecting the protein stability of the other subunits, as well as EMC1 (Fig. 1d, e; the band intensities of the indicated EMC subunit under the different knockdown conditions are quantified in the graph below). These results demonstrate that the EMC4, EMC6, and EMC7 siRNAs used in our experiments specifically decreased the expression of the intended EMC subunit while largely maintaining the stability of the other subunits. The findings that acute depletion of EMC4 did not perturb the stability of EMC1, EMC6, or EMC7, and that transient knockdown of EMC7 did not affect the stability of EMC1, EMC4, or EMC6 are consistent with a recent report demonstrating that acute depletion of EMC4 or EMC7 did not affect the stability of any of the other nine EMC subunits[23]. Hence, our data strengthen the idea that EMC4, EMC6, and EMC7 execute selective and critical functions during SV40 entry.

**EMC4 and EMC7 regulate ER-arrival of SV40 from the cell surface**. What step during SV40 entry might be regulated by EMC4, EMC6, and EMC7? Because EMC1 controls ER-to-cytosol membrane transport of SV40, we asked if this step is also impaired due to loss of EMC4, EMC6, or EMC7. To test this, we used a cell-based, semipermeabilized membrane transport assay previously developed by our lab to monitor cytosol-arrival of SV40 from the ER[17]. In this assay, SV40-infected CV-1 cells were treated with the detergent digitonin that selectively permeabilizes the plasma membrane while leaving internal membranes (including the ER) intact. The samples were centrifuged to generate a supernatant and pellet fraction. The supernatant represents the cytosol fraction, and should therefore comprise of cytosolic proteins as well as SV40 that escaped into the cytosol from the ER. By contrast, the pellet represents the membrane fraction, and should therefore harbor cellular membranes and SV40 that are retained in membranous compartments (such as the ER). To validate the integrity of this fractionation protocol, we immunoblotted both the cytosol and membrane fractions for presence of Hsp90 (a cytosolic marker) and the ER-resident BiP (a membrane marker). Using this fractionation method, we found that depletion of EMC4, EMC6, or EMC7 markedly decreased the SV40 VP1 level in the cytosol (Fig. 2a, top panel, compare lanes 2-4 to 1; the VP1 band intensity in the cytosol is quantified in Fig. 2b), indicating that loss of these individual EMC subunits decreased cytosol-arrival of SV40 from the ER.

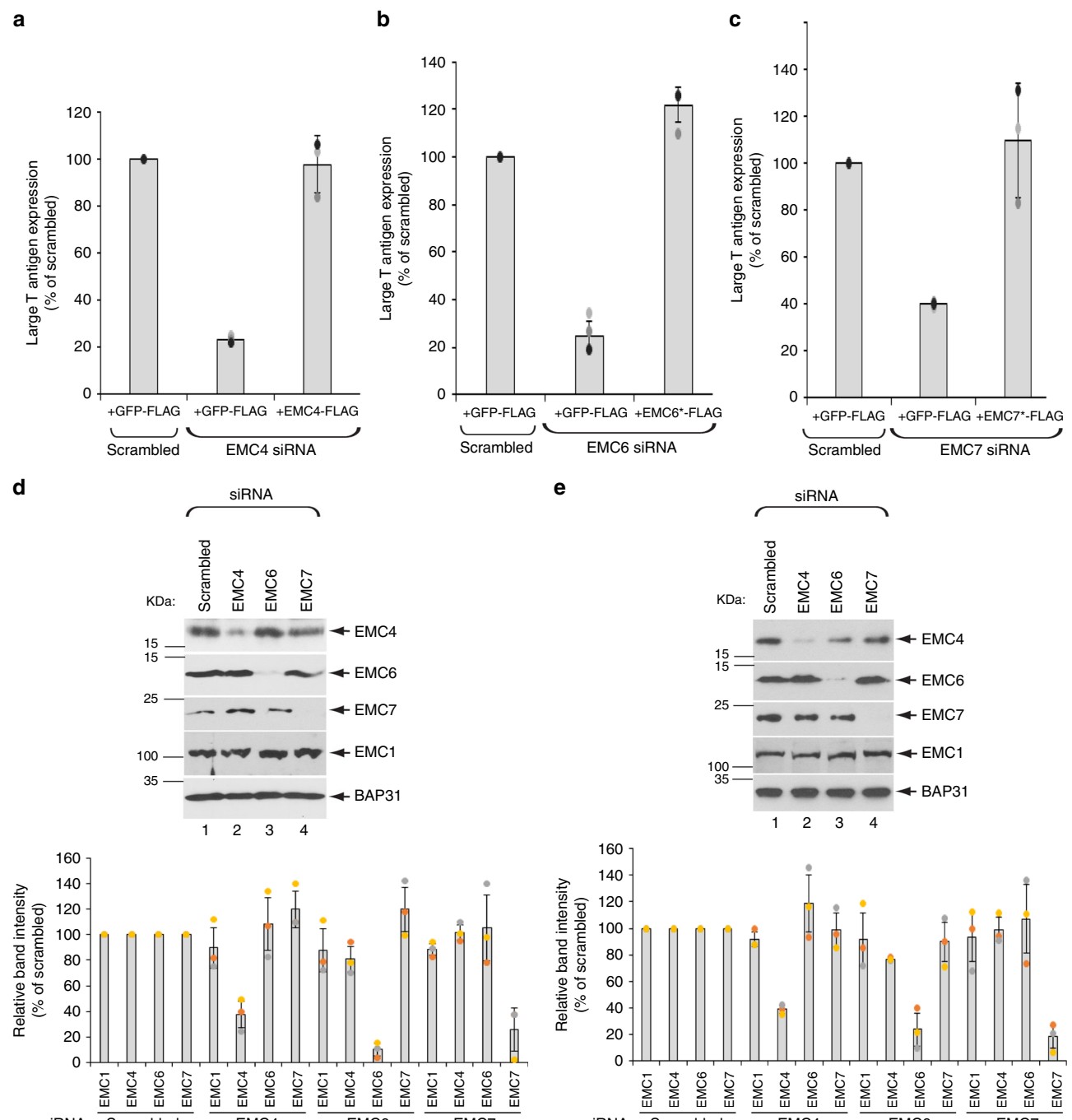

**Fig. 1 EMC4, EMC6, and EMC7 promote SV40 infection. a–c** CV-1 cells were transfected with scrambled or the indicated siRNA, and then with the indicated FLAG-tagged constructs. Cells were then infected with SV40 (MOI ~ 0.5), fixed, and stained with FLAG and large T-antigen antibodies. The percentages of T-antigen-positive cells were determined only in FLAG-expressing cells by using epifluorescence widefield microscopy. Values represent means ± SD from three independent experiments. D-E. CV-1 (**d**) and HEK 293 T (**e**) cells were transfected with the indicated siRNA, and the resulting cell extract was subjected to SDS-PAGE followed by immunoblotting using the indicated antibodies. The indicated EMC band intensities under the different knockdown conditions were quantified in the bottom panel. Values represent means ± SD from three independent experiments. Source data are provided as a Source Data file.

We then assessed if ER-arrival of SV40 from the cell surface was affected by depletion of EMC4, EMC6, or EMC7 using two different approaches. First, we used a previously established biochemical protocol designed to extract ER-localized SV40 from the membrane fraction[17]. Using this method, we found that knockdown of EMC4 or EMC7 but not EMC6 markedly decreased the ER-localized VP1 level (Fig. 2c, bottom panel, compare lane 2 and 4 to 1 and 3; the VP1 band intensity in the ER-localized fraction is quantified in Fig. 2d). These findings suggest that EMC4 and EMC7 regulate ER-arrival of SV40 from the cell surface, while EMC6 likely impacts cytosol entry of the virus from the ER.

Second, to complement this biochemical fractionation approach, we used an imaging-based strategy to verify the role of these EMC subunits during SV40 entry. This approach is based on exposure of the viral internal proteins VP2 and VP3, which

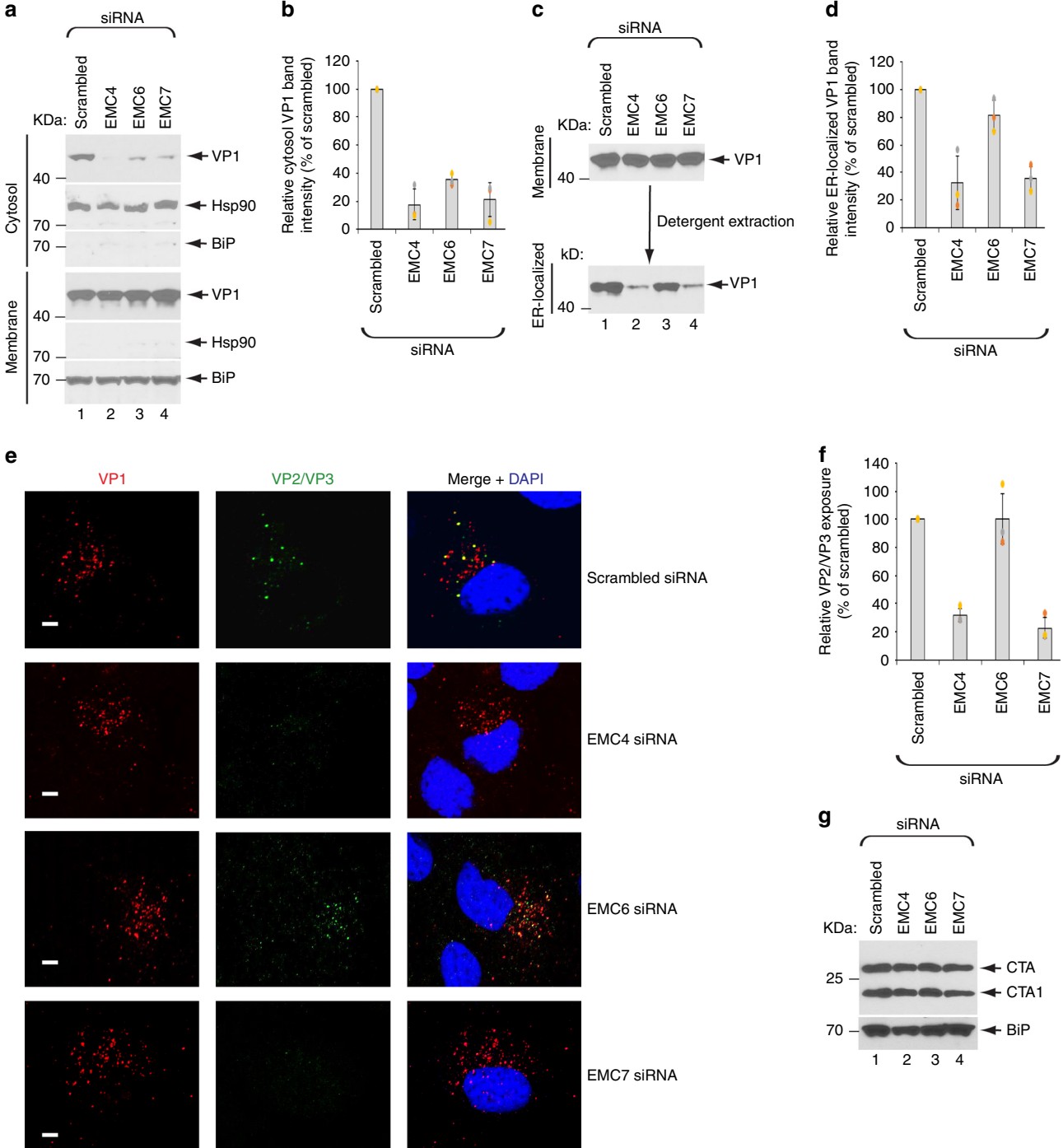

**Fig. 2 EMC4 and EMC7 regulate ER-arrival of SV40 from the cell surface. a** CV-1 cells transfected with the indicated siRNAs were infected with SV40 at MOI ~5, harvested, and subjected to the ER-to-cytosol membrane transport assay (see Materials and Methods). Cytosolic Hsp90 and ER-resident BiP were used as markers for the cytosol and membrane fractions, respectively. **b** Relative VP1 band intensities from the cytosol fraction in A were determined using FIJI/ImageJ. Data are normalized to scrambled siRNA. Values represent the mean ± SD of three independent experiments. **c** To isolate ER-localized SV40, the membrane fraction in A was solubilized in 1% Triton X-100, and the extracted material subjected to SDS-PAGE followed by immunoblotting with the VP1 antibody. **d**. Relative VP1 band intensities of ER-localized SV40 in C were determined using FIJI/ImageJ. Data are normalized to scrambled siRNA. Values represent the mean ± SD of three independent experiments. **e** CV-1 cells transfected with the indicated siRNAs were infected with SV40 (MOI ~0.5) and at 6 h post infection cells were fixed, stained with the indicated antibodies, and analyzed by confocal microscopy. Scale bar = 10 μm. **f** Relative VP2/VP3 exposure from the data in **e** was determined by quantifying VP2/VP3 signals normalized to the VP1 signals by FIJI/ImageJ, and displayed as the percentage of scrambled siRNA-treated sample. The image analysis was done on a single z-plane. Values represent the mean ± SD of three independent experiments. **g** CV-1 cells transfected with the indicated siRNAs were incubated with cholera toxin. Cells were harvested, lysed, and the resulting extract subjected to non-reducing SDS-PAGE and immunoblotted using the indicated antibodies. The experiment was repeated independently three times. Source data are provided as a Source Data file.

occurs only upon successful arrival of SV40 to the ER from the plasma membrane[24]; VP2 and VP3 become exposed in the ER because ER-resident protein disulfide isomerase (PDI) family members reduce or isomerize disulfide bonds in SV40, thereby inducing conformational changes to the viral particle[25,26]. In control cells infected with SV40, VP2/VP3-positive signals can be readily detected (Fig. 2e, row 1). By contrast, in EMC4- and EMC7- (but not EMC6-) depleted cells infected with the virus, a significant decrease in VP2/VP3-positive signals was observed (Fig. 2e, compare rows 2 and 4 to 3; the extent of VP2/VP3 exposure was quantified in Fig. 2f). It is possible that loss of EMC4 or EMC7 globally disrupted all retrograde transport to the ER. However, this is unlikely the case because we found that during entry, ER-arrival of the bacterial toxin cholera toxin (CT) from the cell surface was unaffected by loss of EMC4, EMC6, or EMC7; this is based on the observation that ER-dependent formation of the catalytic CTA1 peptide of CT occurred even in the absence of any of these EMC subunits (Fig. 2g). These findings demonstrate that EMC4 and EMC7 selectively promote arrival of SV40 to the ER from the plasma membrane, and are in complete agreement with the biochemical membrane fractionation studies.

**SV40 accumulates in the late endosome in cells depleted of EMC4 or EMC7.** Upon endocytosis, SV40 reaches the early endosome (EE) and then the late endosome (LE) from where it bypasses the Golgi en route to the ER[15], a pathway similar to other PyVs[13,14,16]. We therefore reasoned that if SV40 cannot efficiently reach the ER in cells depleted of EMC4 or EMC7, the virus is likely trapped in the LE. Colocalization studies was conducted to test this possibility. Specifically, CV-1 cells transfected with scrambled, EMC4, EMC6, or EMC7 siRNA were infected with SV40 and at 6 hpi, fixed and probed with VP1 and Rab7, a cytosolic small GTPase bound to the LE membrane that is commonly used as a LE marker[27]. Cells were then analyzed by confocal microscopy. In control cells, a small level of VP1 signal colocalizes with Rab7 (Fig. 3a, see inset in upper left panels; the VP1-Rab7 colocalization is quantified in Fig. 3b as Mander's Overlap Coefficient). Importantly, the level of VP1-Rab7 colocalization increased in EMC4- and EMC7- but not EMC6-depleted cells (Fig. 3a, compare inset in upper right and lower left panels to lower right panels; quantified in Fig. 3b). These findings suggest that SV40 accumulates in the LE in the absence of EMC4 and EMC7, consistent with the finding that the virus cannot efficiently reach the ER under these compromised conditions.

We conducted similar experiments to assess colocalization of SV40 with the EE marker EEA1. However, no significant increase in VP1-EEA1 colocalization was observed under any of the knockdown conditions (Supplementary Figure 1A; quantified in Supplementary Fig. 1B), suggesting that entrapment of SV40 in the LE due to depletion of EMC4 or EMC7 does not lead to build-up of the virus in the EE compartment.

**EMC4 and EMC7 are in close proximity to Rab7 and bind to LE-associated Rab7.** Our data thus far suggest that EMC4 and EMC7 support LE-to-ER transport of SV40 in order to promote productive infection. How might these EMC subunits accomplish this feat? One possibility is that these ER membrane proteins are in close physical proximity to the LE, potentially stabilizing the LE and ER contact to facilitate delivery of SV40 between these two organelles. To test this idea, we used the proximity-dependent biotin identification (BioID) approach. This technique relies on genetic fusion of the promiscuous biotin ligase (BioID2) to a protein of interest. When this fusion protein is expressed in cells, it will biotinylate endogenous proteins within

10 nm of the fusion protein[28,29]. An advantage of the BioID approach is that it can identify weak or transient interactions[28,29]. Accordingly, we fused Myc-tagged BioID2 to the cytosolic N-terminus of EMC4 (Myc-BioID2-EMC4), and HA-tagged BioID2 to the cytosolic C-terminus of EMC7 (EMC7-BioID2-HA), and used Myc-BioID2 as the negative control construct (Supplementary Fig. 2A). As expected, while Myc-BioID2 is expressed throughout the cell, Myc-BioID2-EMC4 and EMC7-BioID2-HA displayed an ER localization phenotype because these proteins colocalized with the ER transmembrane protein BAP31 (Supplementary Fig. 2B). To test if Myc-BioID2-EMC4 and EMC7-BioID2-HA can biotinylate the prominent LE-associated protein Rab7 GTPase, these constructs and Myc-BioID2 were expressed independently in cells, the resulting extract subjected to precipitation using Streptavidin-conjugated beads to pull down biotinylated proteins, and the precipitated material subjected to immunoblotting. Using this method, we found that endogenous Rab7 (but not the EE marker Rab5) was present in the precipitated material derived from cells expressing Myc-BioID2-EMC4 and EMC7-BioID2-HA but not Myc-BioID2 (Fig. 4a, first panel, compare lanes 2 and 3 to 1). Thus, Myc-BioID2-EMC4, and EMC7-BioID2-HA but not Myc-BioID2 biotinylated Rab7; although Myc-BioID2 did not biotinylate Rab7, it was nonetheless active because it biotinylated numerous cellular proteins (Fig. 4a, second panel, lane 1). These findings indicate that EMC4 and EMC7 are in close physical proximity to Rab7 in cells.

Physical proximity to Rab7 suggests that EMC4 and EMC7 may bind to Rab7. Indeed, we found that immunoprecipitation of EMC4 co-precipitated Rab7 (Fig. 4b, top panel, compare lane 2 to 1), demonstrating that EMC4 binds to Rab7. We next asked if EMC4 selectively interacts with Rab7 that is associated with the LE; Rab7, in its GTP-bound state, associates with the LE[27]. Accordingly, cells were transfected with either the control construct GFP-FLAG, EGFP-tagged wild-type Rab7 (EGFP-WT Rab7), constitutively-active Rab7 (EGFP-Q67L Rab7), or dominant-negative Rab7 (EGFP-N125I Rab7). WT Rab7 cycles between the GTP-GDP states and Q67L Rab7 is locked in the GTP-bound state—both of these proteins associate with the LE[27]. By contrast, N125I Rab7 does not associate with the LE because it exists in the apo form devoid of GTP (or GDP)[27]. Strikingly, immunoprecipitation of EMC4 pulled down only EGFP-WT Rab7 and EGFP-Q67L Rab7 but not EGFP-N125I Rab7 or GFP-FLAG (Fig. 4c, top panel, compare lanes 6 and 8 to lanes 7 and 5). These results suggest that EMC4 interacts with the pool of Rab7 that is associated with the LE. The EMC4-Rab7 interaction does not depend on presence of EMC7 because EMC7 depletion did not perturb binding of EGFP-WT Rab7 to EMC4 (Fig. 4d, top panel, compare lane 3 to 2).

Similar findings were observed for the EMC7-Rab7 interaction. Precipitation of EMC7∗-FLAG (in EMC7-depleted cells) pulled down EGFP-WT Rab7 but not EGFP-N125I Rab7 (Fig. 4e, top panel, compare lane 2 to 3); this interaction was preserved even in the absence of EMC4 (Fig. 4e, top panel, compare lane 4 to 2). Together, these data support the BioID analysis, demonstrating that EMC4 and EMC7 can both interact with the LE-associated Rab7, and that the interactions occur independent of each other.

The ER transmembrane protein protrudin was previously shown to bind directly to Rab7 and facilitates ER-LE contact[30]. We therefore asked if EMC4 or EMC7 also interacts with protrudin, and whether protrudin plays any role in SV40 infection. When endogenous EMC4 was immunoprecipitated, endogenous EMC7 but not protrudin co-precipitated (Supplementary Fig. 3A, compare first and second panels). In addition, knockdown of protrudin (Supplementary Figure 3B, first panel, compare lane 2 to 1) did not impair SV40 infection, in contrast to depletion of EMC4 (Supplementary Fig. 3C). These data suggest

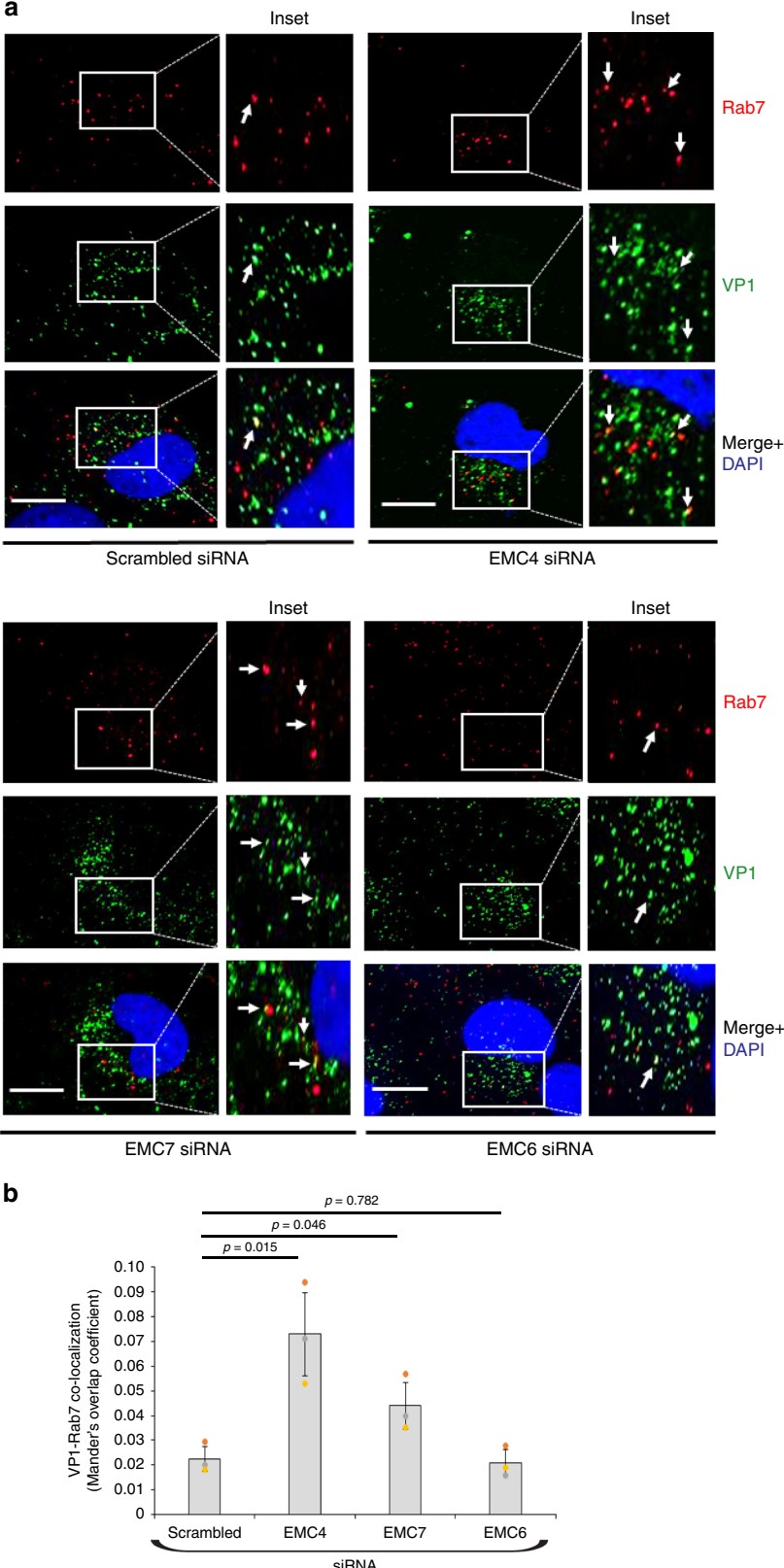

**Fig. 3 SV40 accumulates in the late endosome in cells depleted of EMC4 or EMC7. a** CV-1 cells transfected with indicated siRNA were infected with SV40 (MOI ~10), fixed at 6 h post infection, and stained with specific antibodies followed by confocal microscopy. DAPI positions the nucleus. Scale bar = 10 μm. **b** VP1-Rab7 colocalization was quantified as Mander's Overlap Coefficient using FIJI/ImageJ coloc2 plugin. The image analysis was done on a single z-plane. Values represent the mean ± SD of three independent experiments and at least fifty infected cells were analyzed for every conditions during each independent experiments. Unpaired Student two-tailed $t$-test was used to determine statistical significance. Source data are provided as a Source Data file.

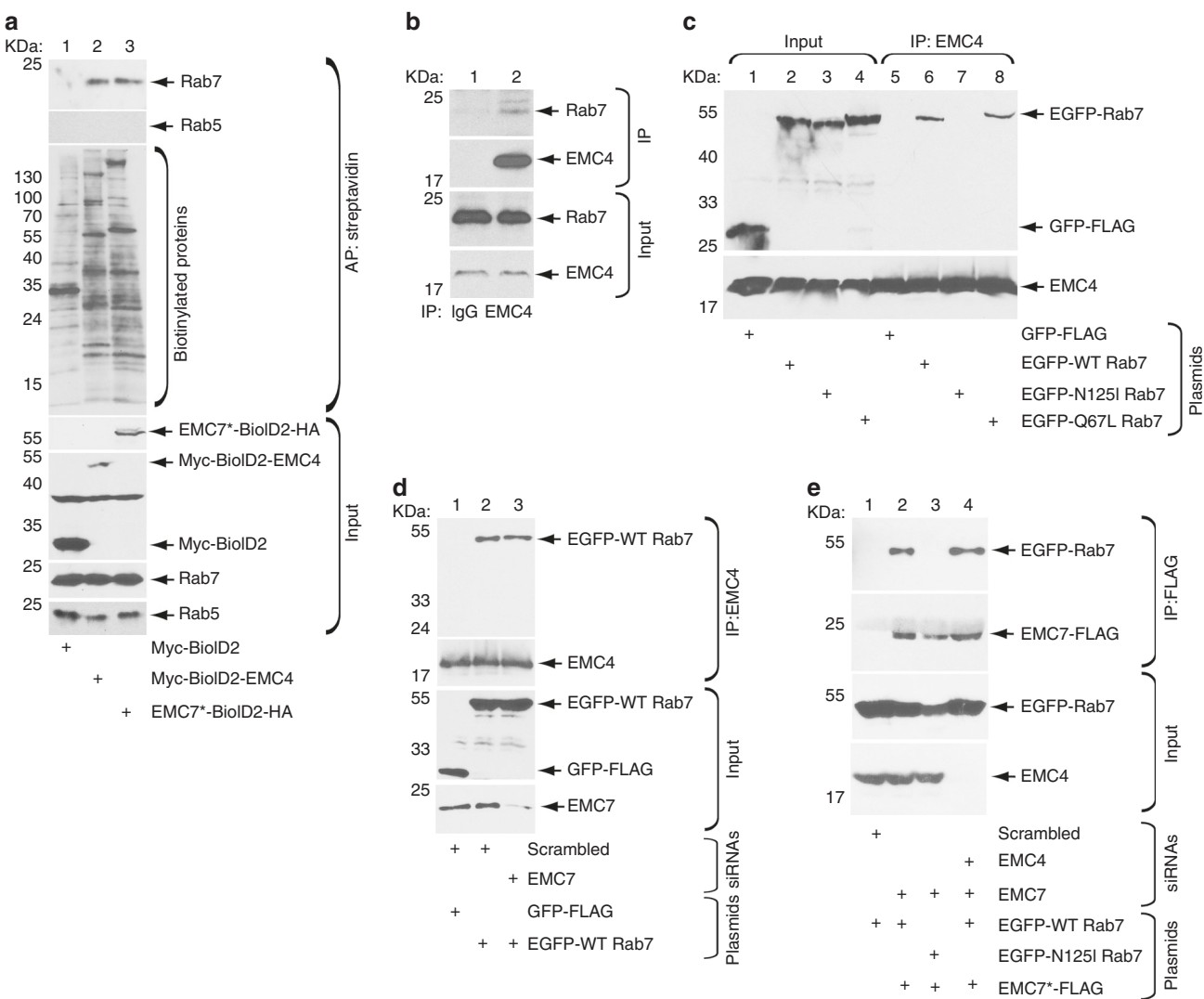

**Fig. 4 EMC4 and EMC7 are in close proximity to Rab7 and bind to LE-associated Rab7. a** HEK 293 T cells expressing either Myc-BioID2, Myc-BioID2-EMC4, or EMC7-BioID2-HA were treated with biotin, and lysed with a RIPA buffer (containing 1% SDS). The resulting extract was subjected to affinity purification with Streptavidin C1 beads and the bound material eluted by a biotin-containing elution buffer. The input and eluted materials were subjected to SDS-PAGE and immunoblotted with indicated antibodies or probed with Streptavidin-HRP (to detect the total biotinylated proteins in each sample). **b** HEK 293 T cells were lysed and the resulting extract was incubated with either the control IgG or EMC4 antibody. The precipitated material was subjected to SDS-PAGE followed by immunoblotting with the indicated antibodies. **c** HEK 293 T cells transfected with the indicated plasmid were lysed, the resulting extract was incubated with EMC4 antibody, and the precipitated material subjected to SDS-PAGE and immunoblotting with the indicated antibodies. **d–e** HEK 293 T cells transfected with the indicated siRNA and DNA construct were lysed and the resulting extract was incubated with either an EMC4 (**d**) or FLAG antibody-conjugated agarose beads (**e**). The precipitated material was subjected to SDS-PAGE and immunoblotted with the indicated antibodies. All experiments were independently repeated three times. Source data are provided as a Source Data file.

that Rab7's interaction with EMC4 and EMC7 is distinct from the previously reported Rab7-protrudin interaction.

**EMC4 and EMC7 support colocalization of the ER and LE**. Our analyses demonstrating close proximity and physical association between EMC4-EMC7 and LE-associated Rab7, as well as our finding showing defective ER-arrival of SV40 under EMC4 or EMC7 knockdown condition, raise the possibility that these EMC subunits tether the ER to the LE in order to support productive intracellular transport of SV40. In this scenario, we envision contact between these two organelles should be disrupted in the absence of EMC4 and EMC7. We used super-resolution structured illumination microscopy (SIM) to test this possibility. Indeed, while colocalization between the ER (BAP31) and LE (STARD3) can be detected in control SV40-infected cells expressing GFP11-STARD3-FLAG (Fig. 5a, top panels, see

colocalized pixels in white), the extent of colocalization decreased when EMC4 and EMC7 were depleted (Fig. 5a, bottom panels, see colocalized pixels in white; colocalization was quantified by the Mander's Overlap Coefficient in Fig. 5b). Of note, GFP11 is appended to STARD3-FLAG because it is needed for a subsequent experiment (see Fig. 6). As complementary quantitative readouts, we also compared the scatter plot (Supplementary Fig. 4A) and colocalized pixel map (constant intensity) (Supplementary Fig. 4B) corresponding to images derived from scrambled and EMC4/EMC7-depleted cells. Together, these findings suggest that EMC4 and EMC7 support colocalization between the ER and LE.

**EMC4 and EMC7 mediate ER-LE contact**. Thus far, the BioID and co-precipitation strategies revealed that EMC4 and EMC7 are in physical proximity to the LE-associated Rab7, while the super-

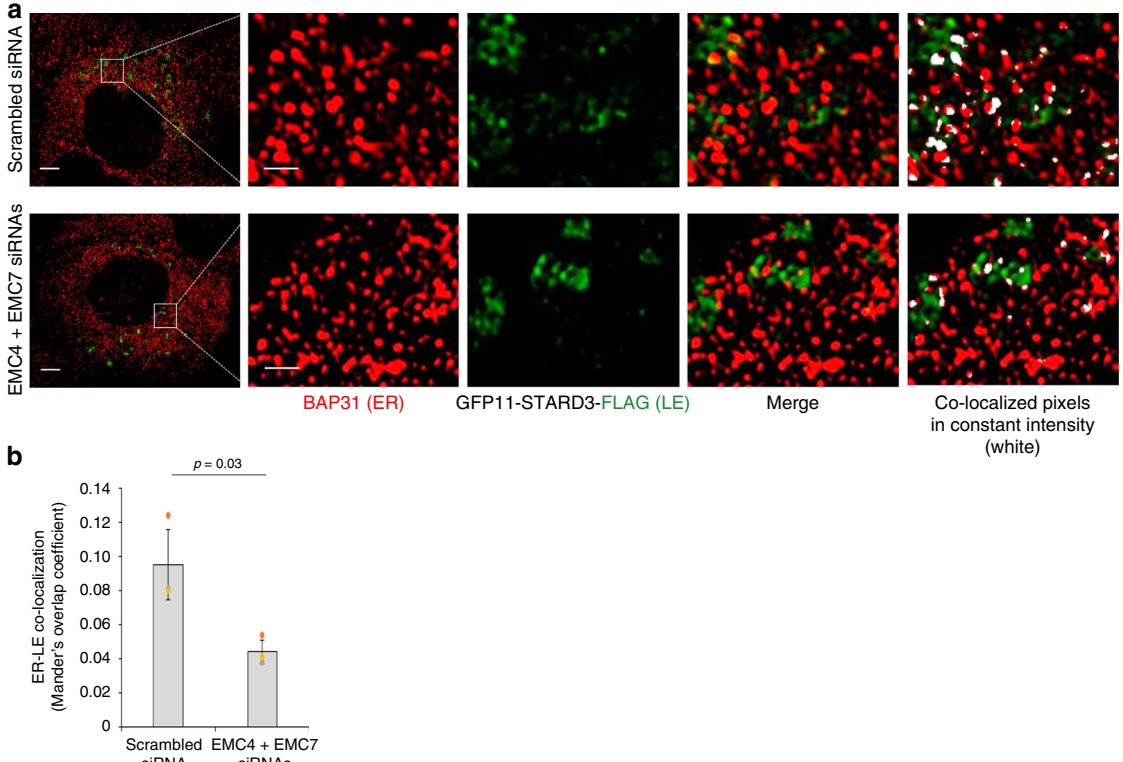

**Fig. 5 EMC4 and EMC7 support colocalization of the ER and LE. a** GFP11-STARD3-FLAG-expressing CV-1 cells transfected with scrambled or EMC4 +EMC7 siRNAs were infected with SV40, fixed at 5 h post infection, and stained with BAP31 (ER marker) and FLAG (LE marker) antibodies. Cells were analyzed by structured illumination microscopy (SIM). Images with colocalized pixels were generated with FIJI/ImageJ colocalization threshold plugin. Scale bar in the first column = 5 µm, scale bar in the second column = 1 µm. **b** ER-LE colocalization in A was quantified by the Mander's Overlap Coefficient using the FIJI/ImageJ coloc2 plugin. The image analysis was done on a single z-plane. Values represent the mean ± SD from three independent experiments, and at least ten FLAG-expressing cells were analyzed for every condition during each independent experiment. Unpaired Student two-tailed *t*-test was used to determine statistical significance. Source data are provided as a Source Data file.

resolution microscopy approach demonstrated that these two EMC subunits mediate colocalization of ER and LE. Based on these observations, we asked if EMC4 and EMC7 promote ER-LE contact. We used a previously described split-GFP approach[31] to test this possibility. The LE transmembrane protein STARD3 was tagged with GFP11 (at its N-terminus) and FLAG (at its C-terminus), generating GFP11-STARD3-FLAG. Additionally, GFP (1-10) was appended to an unrelated ER transmembrane protein B14 [GFP (1-10)-B14]. As both portions of GFP appended to the different membrane proteins are displayed on the cytosolic side of their respective membranes, GFP fluorescence is expected to be reconstituted only if the two organelles—LE and ER—come close and form a membrane contact site (Fig. 6a).

As controls, we found that no GFP signal is observed if cells expressed either GFP11-STARD3-FLAG or GFP (1-10)-B14 alone (Fig. 6b, top and middle rows). By contrast, distinct GFP signal is observed in cells expressing both GFP11-STARD3-FLAG and GFP (1-10)-B14 (Fig. 6b, bottom row). We reasoned that if selective ER membrane proteins tether the ER to the LE, depletion of these tethering proteins should decrease the GFP intensity under this experimental setup. To validate this split-GFP assay as a bonafide strategy for assessing ER-LE contact, we depleted the ER membrane protein VAPA/B and found that the GFP signal was indeed decreased (Supplementary Fig. 5); VAPA/ B were previously shown to be ER-LE tethering proteins[32–34]. Strikingly, depletion of EMC4 and EMC7 also decreased the GFP intensity (Fig. 6c; quantified in Fig. 6d), strongly suggesting that these EMC subunits mediate LE-ER contact.

We used electron microscopy (EM) to strengthen this idea. By standard transmission EM (TEM), we find evidence of SV40 particles within the LE that appears to contact the ER in SV40-infected cells (Supplementary Fig. 6A). We then used immuno-EM to determine if EMC4 and EMC7 might be at these contact sites. In cells depleted of EMC4 but expressing EMC4-FLAG, FLAG-positive signal can be found at (or proximal) to the LE vesicle, as marked by Rab7 (Supplementary Fig. 6B) or STARD3 (Supplementary Fig. 6C). Similarly, in cells depleted of EMC7 but expressing EMC7-FLAG, FLAG-positive signal can be observed to be close to the LE organelle, as indicated by Rab7 (Supplementary Fig. 6D) or STARD3 (Supplementary Fig. 6E); intriguingly, a SV40 particle can be seen in the Rab7-positive LE making contact with the EMC7-positive ER (Supplementary Fig. 6D). These immuno-EM data thus support the co-IP, BioID, super-resolution, and split-GFP findings, suggesting that EMC4 and EMC7 act as tethers to support ER-LE contact.

**Specific domains of EMC4 and EMC7 mediate Rab7-binding essential for SV40 ER-arrival and infection.** We next sought to identify specific domains within EMC4 and EMC7 that are essential for interaction with Rab7. Based on primary amino acid sequence, EMC4 is predicted to be a two-pass transmembrane protein in which its N- and C-terminal ends are both oriented toward the cytosol[35]. In this topology, amino acids 1-85 represent the N-terminal extension facing the cytosol. Within this sequence, a protein–protein interaction domain called the low complexity

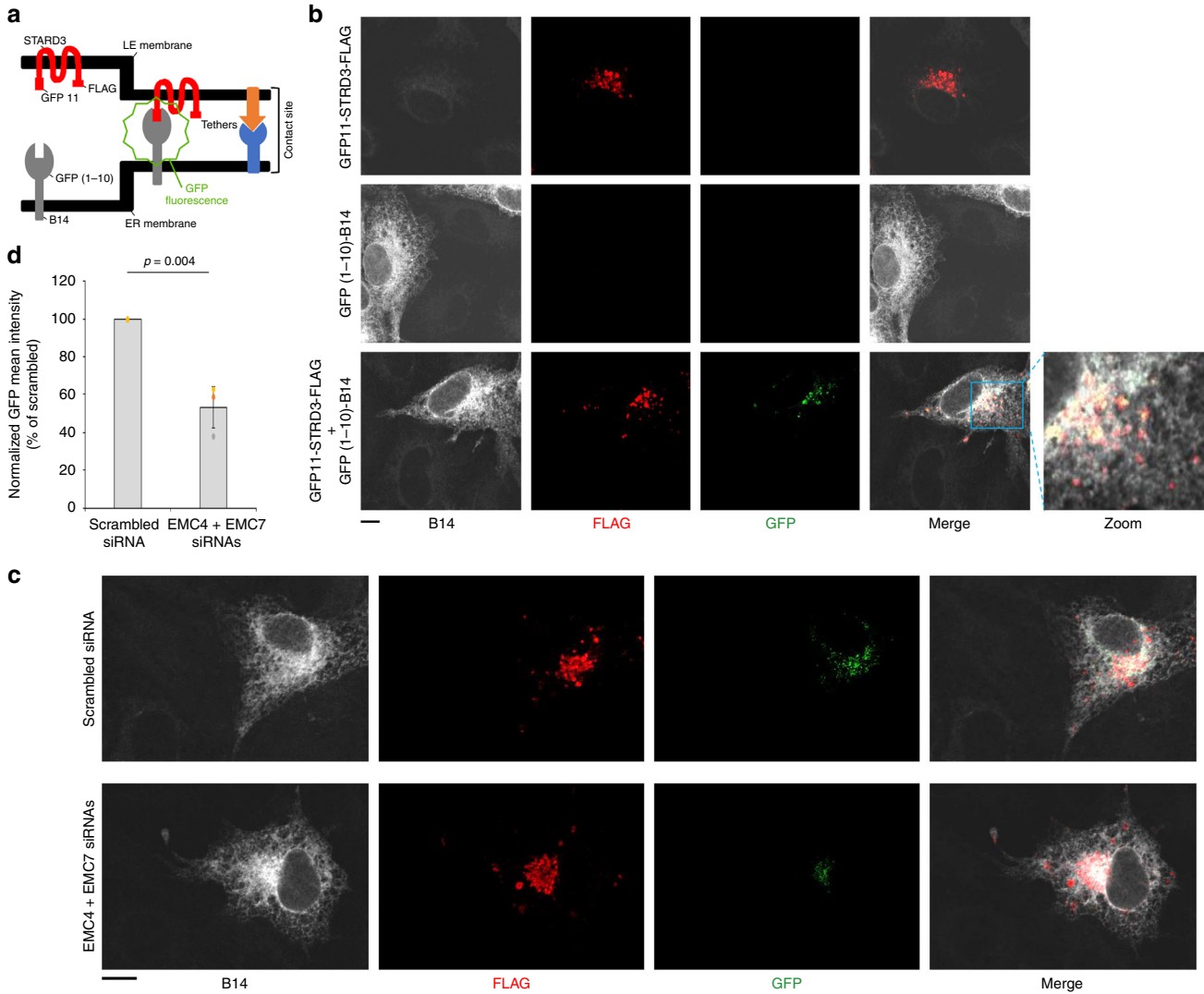

**Fig. 6 EMC4 and EMC7 mediate ER-LE contact. a** Graphical representation of the split-GFP assay for detecting ER-LE contacts. **b** CV-1 cells transfected with the indicated construct(s) were fixed, stained with B14 and FLAG antibody, and analyzed under confocal microscope for GFP signal. This experiment was independently repeated three times. Scale bar = 10 μm. **c** CV-1 cells co-expressing GFP11-STARD3-FLAG and GFP (1-10)-B14 and transfected with either scrambled or EMC4+EMC7 siRNAs were fixed, stained with B14 and FLAG antibody, and analyzed under confocal microscope for GFP signal. Scale bar = 10 μm. **d** Mean GFP signal intensity from **c** was normalized with FLAG signal and quantified by FIJI/ImageJ. The image analysis was done on a single z-plane. Values represent means ± SD from three independent experiments. Unpaired Student two-tailed t-test was used to determine statistical significance. Source data are provided as a Source Data file.

region (LCR) located at positions 21-40 can be identified based on pfam (Fig. 7a). Importantly, the cytosolic LCR of protrudin was previously shown to bind directly to Rab7 in order to facilitate ER-LE contact[30]. We therefore asked if an N-terminal FLAG-tagged EMC4 construct missing its first 50 amino acids (FLAG-Δ50 EMC4), and therefore lacking the LCR, binds to Rab7. The corresponding N-terminal FLAG-tagged WT EMC4 construct (FLAG-EMC4) was also generated. When transfected, FLAG-Δ50 EMC4 colocalizes with BAP31, similar to FLAG-EMC4 (Supplementary Fig. 7A, left panels). Additionally, precipitation of FLAG-Δ50 EMC4 (but not the control GFP-FLAG) pulled down the other EMC subunits EMC1, EMC6, and EMC7 with the same efficiency as FLAG-EMC4 (Supplementary Fig. 7B, compare lane 3 to 2). These findings indicate that FLAG-Δ50 EMC4 properly interacts with the other EMC subunits in the ER membrane. However, FLAG-Δ50 EMC4 did not efficiently bind to EGFP-Rab7 when compared to FLAG-EMC4 (Fig. 7b, top panel, compare lane 2 to 1). This result demonstrates that the first 50 amino

acids of EMC4 containing the LCR is responsible for Rab7 interaction.

To assess the functional consequence of deleting the LCR from EMC4 during SV40 entry, we asked if expression of FLAG-Δ50 EMC4 can restore the block in virus infection due to knockdown of EMC4. In cells depleted of EMC4, whereas expression of FLAG-EMC4 restored virus infection (Fig. 7c, compare third to second bar), similar to EMC4-FLAG (Fig. 1a), expressing FLAG-Δ50 EMC4 did not (Fig. 7c, compare fourth to second bar). Hence, the N-terminal 50 amino acids of EMC4 is required during SV40 infection. This region of EMC4 is also critical for supporting ER-arrival of SV40 from the cell surface because only expression of FLAG-EMC4 but not FLAG-Δ50 EMC4 restored VP2/VP3 exposure in cells depleted of EMC4 (Fig. 7d, compare third to fourth bar). Thus the N-terminal cytosolic domain of EMC4 that harbors a unique LCR essential for Rab7 binding is functionally important to promote ER-arrival of SV40 leading to productive infection.

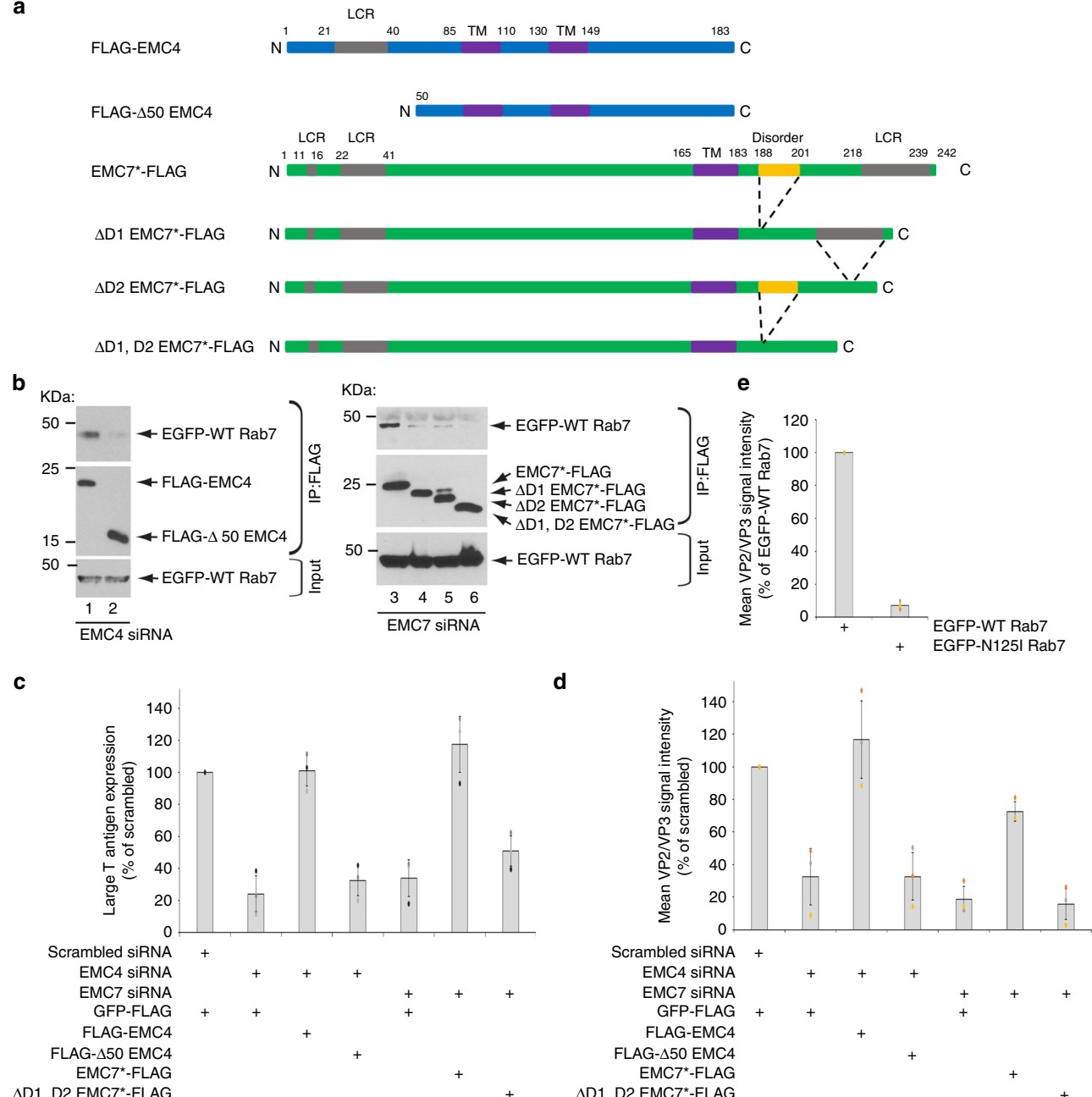

**Fig. 7 Specific domains of EMC4 and EMC7 mediate Rab7-binding essential for SV40 ER-arrival and infection. a** Diagram of wild-type and mutant EMC4 and EMC7. **b** EMC4 (lane 1-2) or EMC7 (lane 3–6) siRNA-treated HEK 293 T cells transfected with the indicated plasmid were lysed, the resulting extract incubated with FLAG antibody-conjugated agarose beads, and the precipitated material subjected to SDS-PAGE and immunoblotting. This experiment was independently repeated three times. **c** CV-1 cells were transfected with scrambled or the indicated siRNAs prior to transfection with the indicated FLAG-tagged constructs. Cells were then infected with SV40 (MOI ~ 0.5), fixed, and stained with FLAG and large T-antigen antibodies. The percentages of T-antigen-positive cells were determined in only FLAG-expressing cells by using epifluorescence widefield microscopy. Values represent means ± SD from three independent experiments. **d** CV-1 cells were transfected with scrambled or the indicated siRNAs prior to transfection with the indicated FLAG-tagged constructs. Cells were then infected with SV40 (MOI ~ 2), fixed, and stained with FLAG and VP2/VP3 antibodies. The mean VP2/VP3 signal intensity was determined by FIJI/ImageJ software in only FLAG-expressing cells by using epifluorescence widefield microscopy. Values represent means ± SD from three independent experiments. **e** CV-1 cells were transfected with the indicated EGFP-tagged construct. Cells were then infected with SV40, fixed, and stained with VP2/VP3 antibodies. The mean VP2/VP3 signal intensity was determined by FIJI/ImageJ software in only EGFP-expressing cells by using epifluorescence widefield microscopy. Values represent means ± SD from three independent experiments. Source data are provided as a Source Data file.

A similar approach was used to identify domains within EMC7 that engage Rab7. As a single-pass type I transmembrane protein, EMC7 orients its C-terminus (amino acids 184-242) towards the cytosol[35]. Based on pfam, the protein–protein interaction "disordered" domain (located at positions 188-201) and a LCR (located at positions 218-239) can be found within the C-terminal cytosolic domain of EMC7 (Fig. 7a). Hence, in addition to wild-type EMC7∗-FLAG, we generated a truncated EMC7 contruct

lacking either the disordered domain (ΔD1 EMC7∗-FLAG), the LCR (ΔD2 EMC7∗-FLAG), or both (ΔD1, D2 EMC7∗-FLAG) (Fig. 7a).

As control experiments, we found that all of the truncated EMC7 constructs colocalize with BAP31 (Supplementary Fig. 7A, right panels), and bind to EMC1, EMC4, and EMC6 (Supplementary Fig. 7B, lanes 6-8), demonstrating that the truncated EMC7 proteins interact with the other EMC subunits in the ER membrane. However, the truncated EMC7 constructs all displayed defect in binding to EGFP-Rab7 when compared EMC7∗-FLAG (Fig. 7b, top panel, compare lane 4-6 to 3), with ΔD1, D2 EMC7∗-FLAG showing the strongest defect. Strikingly, whereas EMC7∗-FLAG largely restored SV40 infection (Fig. 7c, compare 6th to 5th bar) and ER-arrival (Fig. 7d, compare 6th to 5th bar) in EMC7-depleted cells, ΔD1, D2 EMC7∗-FLAG did not (Fig. 7c, compare 7th to 5th bar; Fig. 7d, compare 7th to 5th bar). We conclude that the disordered domain and LCR harbored within EMC7's cytosolic domain mediate Rab7-binding required for ER-arrival of SV40 enabling productive infection.

Because EMC4 and EMC7 can only interact with LE-associated Rab7 (Fig. 4c, Fig. 4e), we asked whether expression of N125I Rab7, the form of Rab7 that cannot associate with the LE, might block ER-arrival of SV40 from the LE. Indeed, using VP2/3 exposure as a readout of ER-arrival, our results showed that expression of EGFP-N125I Rab7 (but not EGFP-WT Rab7) robustly decreased SV40 arrival to the ER from the LE (Fig. 7e). Thus, EMC4 and EMC7 must interact with LE-associated Rab7 to support proper ER-arrival of SV40 from the LE. In sum, these findings suggest that EMC4 and EMC7 function as molecular tethers, juxtaposing the ER to the LE to enable efficient transport of SV40 from the LE to the ER essential for successful infection.

**EMC4 and EMC7 bind to Stx18 which promotes delivery of SV40 to the ER**. In addition to tethering the LE to the ER, we probed how EMC4 and EMC7 might play a further role in delivering SV40 from the LE to the ER. We envision that SV40 reaches the ER when a vesicle (likely derived from the LE) harboring the viral particle fuses with the ER membrane; this membrane fusion event would de facto deliver the virus into the ER lumen. Vesicular fusion with the ER is mediated by ER-resident SNARE proteins[36]. In fact, in a whole-genome siRNA screen, the ER transmembrane SNARE protein syntaxin18 (Stx18) was found to be essential in endosome-to-ER delivery of the human BK PyV[16]. We therefore asked if EMC4 and EMC7 bind to Stx18. Immunoprecipitation of endogenous Stx18 pulled down endogenous EMC4 and EMC7 (Fig. 8a, first and second panels), demonstrating that Stx18 binds to these EMC subunits. Functionally, knockdown of Stx18 (Fig. 8b, first panel, compare lane 2 to 1) markedly blocked SV40 infection (Fig. 8c), ER-to-cytosol transport (Fig. 8d, e), and importantly, ER-arrival from the cell surface (Fig. 8f, g). These results indicate that EMC4 and EMC7 not only tether the LE via binding to Rab7, but they also interact with the ER-resident SNARE Stx18 that promotes delivery of SV40 to the ER.

## Discussion

Navigating the complex endomembrane system of the host cell during virus entry is a daunting task. In the case of the nonenveloped polyomavirus SV40, upon reaching the LE after endocytosis, the virus must avoid the lysosome and instead be sorted to the ER to successfully cause infection. How this step is executed remains unclear. This manuscript reveals the molecular basis by which EMC4 and EMC7, two members of the ER membrane complex (EMC), support LE-to-ER transport of SV40.

Acute knockdown of EMC4 or EMC7 was previously shown to preserve the stability of the other nine EMC subunits[23]. Consistent with this, we found that acute depletion of EMC4 did not significantly affect the level of EMC1, EMC6, and EMC7, nor did acute depletion of EMC7 markedly alter the level of EMC1, EMC4, and EMC6. Establishing these conditions enabled us to assess the subunit-specific functions of EMC4 and EMC7 during SV40 infection. Strikingly, we found that depletion of EMC4 or EMC7 blocked SV40 infection because the virus cannot reach the ER from the LE, a critical infection step; under this compromised condition, SV40 is trapped in the LE as expected. The impaired ER-arrival of SV40 is not due to global disruption of all retrograde transport processes because ER-arrival of CT remain intact in the absence of EMC4 or EMC7. Because CT is sorted to the ER from the Golgi[37] whereas SV40 is targeted to the ER from the LE[13–16], we postulate that EMC4 and EMC7 support a specific event during LE-ER transport.

Based on this hypothesis, we posited that EMC4 and EMC7 might be in close physical proximity to the LE, and found using the BioID strategy that this was indeed the case. To provide a molecular explanation for this observation, our analysis via the coimmunoprecipitation approach showed that EMC4 and EMC7 interacted with the LE-associated Rab7 GTPase. Not surprisingly, by high-resolution SIM, colocalization between the ER and LE was disrupted in cells depleted of EMC4 and EMC7. Using a split-GFP approach, we further demonstrated that EMC4 and EMC7 promote ER-LE contact, and our immuno-EM analysis suggests the presence of EMC4 and EMC7 at the ER-LE contact. These collective data reveal that EMC4 and EMC7 act as molecular tethers, likely stabilizing membrane contact between ER and LE.

LE-ER contact is an area of intense investigation. In this context, the ER membrane protein protrudin was shown to facilitate LE-ER contact during neurite protrusion and outgrowth[30]. However, our findings showed that EMC4 and EMC7 do not interact with protrudin, and neither did the knockdown of protrudin affected SV40 infection. Thus the EMC4/EMC7-dependent contact with the LE is likely distinct from the protrudin-mediated LE contact. Protrudin contains a cytosolic LCR that binds directly to Rab7[30]. We therefore asked if the LCR in the cytosolic domain of EMC4 and EMC7 mediated Rab7-binding. Our results demonstrated that an EMC4 deletion mutant lacking the LCR cannot bind to Rab7, and an EMC7 deletion mutant lacking its LCR (and the disordered domain) similarly failed to engage Rab7. Importantly, this LCR in these EMC subunits executed a critical role in delivery of SV40 to the ER and infection. These findings further support the idea that the LCR-dependent interaction with Rab7 serves an essential function, enabling EMC4 and EMC7 to recruit the LE in order to facilitate inter-organelle transport of SV40. We note that as only a low level of endogenous Rab7 binds to EMC4, it is possible that the Rab7-EMC4 interaction may be indirect and supported by one of the Rab7 effectors or complexes.

How might EMC4/EMC7-dependent tethering of the LE to the ER support SV40 sorting from LE to ER? One possibility is that SV40 in the LE buds off into a vesicle via a membrane fission step; in fact, ER-LE contact is thought to define a site of endosome fission[38,39]. The newly-generated virus-containing vesicle in turn fuses with the ER membrane, delivering the viral particle into the ER lumen. Vesicular fusion with the ER is likely mediated by an ER-resident SNARE fusion protein[36]. Indeed, we found that Stx18, an ER membrane SNARE essential in endosome-to-ER delivery of BK PyV[16], binds to EMC4 and EMC7, and also plays an important role in sorting SV40 to the ER to promote infection. Hence, the coincident interactions of EMC4 and EMC7 with Rab7 and Stx18 may enable these EMC subunits to coordinate LE-ER tethering with a Stx18-

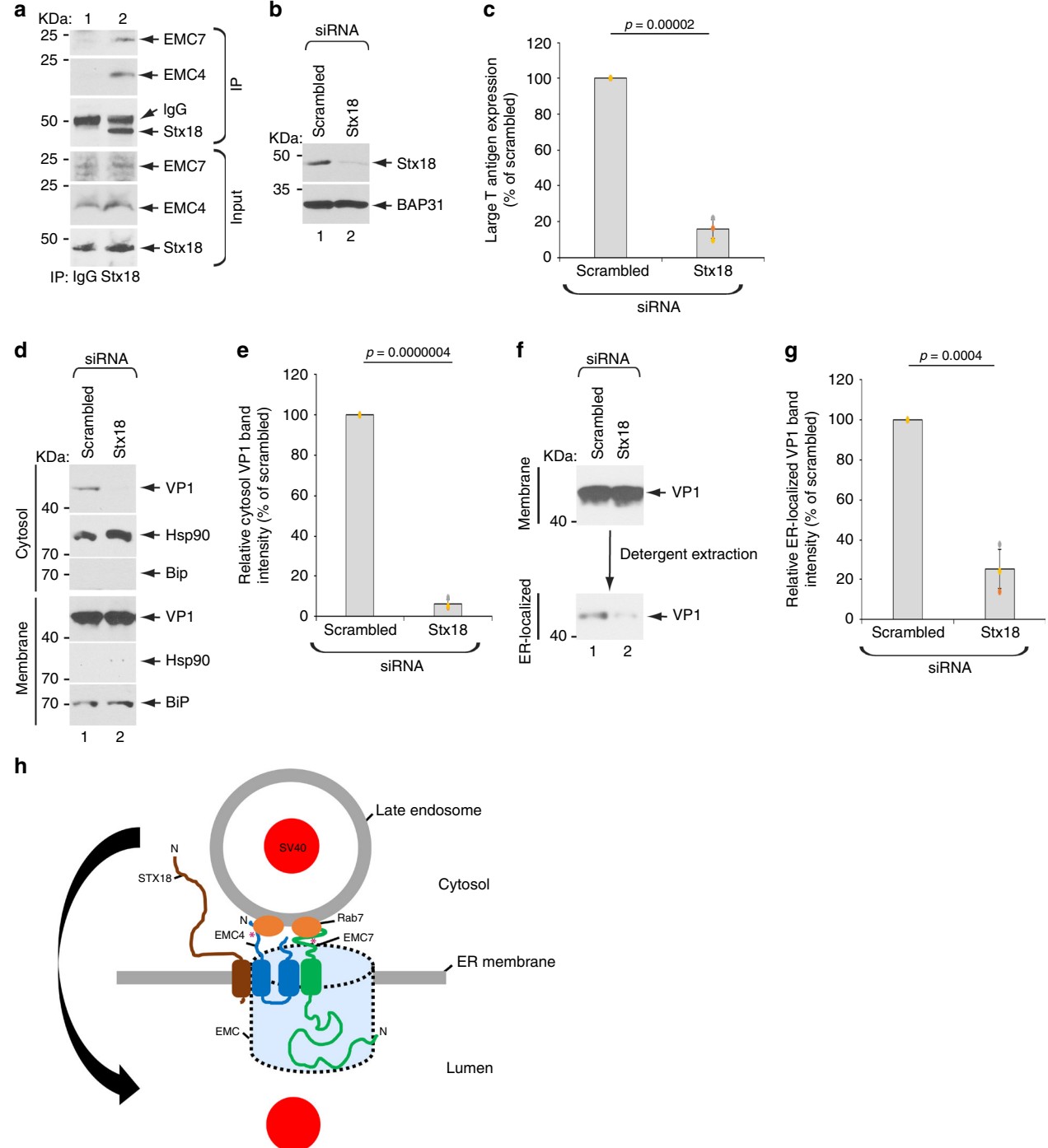

dependent membrane fusion event that delivers LE-localized SV40 to the ER.

Since its initial discovery in 2009, the cellular function of the EMC has remained mysterious[40]. However, recent reports revealed that the EMC acts as a chaperone, promoting the biogenesis of a limited number of multi-pass transmembrane proteins[23,41–45]. This ER membrane complex was also shown to function as an insertase, driving the membrane insertion of select tail-anchored proteins[46]. However, because these studies were conducted using a CRISPR-dependent knockout approach in which all EMC subunits were destabilized, the precise EMC subunits responsible for these reactions are unknown. Intriguingly, the EMC has also been suggested to function as a scaffold,

connecting the ER to the mitochondria in order to promote phospholipid transfer from the ER to the mitochondria[47]. This idea is analogous to our proposal that the EMC (i.e., EMC4 and EMC7) can act as molecular tethers, linking the ER to the late endosome to promote virus transport between these two organelles. Whether this LE-ER tethering activity of the EMC is used for inter-organelle transport of cellular cargos is likely to be a fascinating area for future investigation.

## Methods

**Cells**. CV-1 (male), COS-7(male), and HEK 293 T (male) cells were obtained from ATCC, cultured at 37 °C under 5% $CO_2$ in complete DMEM (cDMEM), containing 10% fetal bovine serum, 10 U/ml penicillin, and 10 µg/ml streptomycin (Gibco,

**Fig. 8 EMC4 and EMC7 bind to Stx18 which promotes delivery of SV40 to the ER. a** HEK 293 T cells were lysed and the resulting extract subjected to immunoprecipitation using either a mouse control IgG or Stx18 antibody. The precipitated material was subjected to SDS-PAGE followed by immunoblotting with the indicated antibodies. This experiment was independently repeated three times. **b** CV-1 cells transfected with the scrambled or Stx18 siRNA were lysed and the resulting extract subjected to SDS-PAGE followed by immunoblotting with the indicated antibodies. This experiment was independently repeated three times. **c** CV-1 cells transfected with the scrambled or Stx18 siRNA were infected with SV40 and the extent of infection was assessed as in Fig. 1. Values represent the mean ± SD of three independent experiments. Unpaired Student two-tailed *t*-test was used to determine statistical significance. **d** CV-1 cells transfected with the indicated siRNAs were infected with SV40 at MOI ~ 5, harvested, and subjected to the ER-to-cytosol membrane transport assay (see Materials and Methods). Cytosolic Hsp90 and ER-resident BiP were used as markers for the cytosol and membrane fractions, respectively. **e** Relative VP1 band intensities from the cytosol fraction from D were determined using FIJI (NIH). Data are normalized to scrambled siRNA. Values represent the mean ± SD of three independent experiments. Unpaired Student two-tailed *t*-test was used to determine statistical significance. **f** To isolate ER-localized SV40, the membrane fraction in **d** was solubilized in 1% Triton X-100, and the extracted material subjected to SDS-PAGE followed by immunoblotting with the VP1 antibody. **g** Relative VP1 band intensities of ER-localized SV40 in **f** were determined using FIJI (NIH). Data are normalized to scrambled siRNA. Values represent the mean ± SD of three independent experiments. Unpaired Student two-tailed *t*-test was used to determine statistical significance. **h** A model depicting how EMC4 and EMC7 act as molecular tethers to juxtapose the ER to the LE. In this model, EMC4 (via its cytosolic LCR) and EMC7 (via its cytosolic LCR and disordered domain) bind to the LE-associated Rab7 directly (or potentially indirectly via other factors) (*), thereby bringing close proximity the ER and the LE. We propose that this juxtaposition supports the efficient transport of SV40 from the LE to the ER, a critical virus infection step. We further propose that the coincident interactions of EMC4 and EMC7 with Rab7 and Stx18 may enable these EMC subunits to coordinate LE-ER tethering with a Stx18-dependent membrane fusion step that conveys LE-localized SV40 to the ER. Source data are provided as a Source Data file.

Grand Island, NY). Dulbecco's modified Eagle's medium (DMEM), 0.25% trypsin-EDTA, HyClone FetalClone III (FC) were purchased from Thermo Fisher Scientific (Waltham, MA). Details of the cell lines are listed in Supplementary Table 2.

**siRNA transfection.** All Star Negative purchased from Qiagen (Valencia, CA) was used as the control siRNA (labeled as scrambled). Pre-designed siRNAs against EMC4 (ID:140362), EMC6 (ID: 33278), EMC7 (ID: 28005), and protrudin (ZFYVE27) (ID: 149130) were purchased from Thermo Fisher Scientific (Waltham, MA) while siRNA pool against syntaxin18 (Stx18) (Cat # L-020624-01-0005)[16] was obtained from Dharmacon (Lafayette, CO). VAPA and VAPB siRNA were generated from Sigma (St Louis, MO) using 5'- GCUACAGCCCUUUGACUAU −3' and 5'- CCGACAGACCGAAAUGUGU −3' sequences, respectively. Using Lipofectamine RNAiMAX (Thermo Fisher Scientific), 30 nM of EMC4 siRNA, 20 nM of EMC6 siRNA, 10 nM of EMC7, or 50 nM of protrudin, VAPA, or VAPB siRNA were reverse transfected into CV-1 or HEK 293 T cells. Infection and biochemical assays were carried out 48 h post transfection.

**Plasmid constructs.** Myc-BioID2 and BioID2-HA plasmids were obtained from Addgene (Catalog number 74223, 74224 respectively) and used to generate Myc-BioID2-EMC4 and EMC7-BioID2-HA constructs using primer sequences listed in Supplementary Table 1. EGFP-WT Rab7, EGFP-N125I Rab7, and EGFP-Q67L Rab7 constructs were kind gifts from Joel Swanson (University of Michigan). Split-GFP sequences were provided by Kristen Verhey (University of Michigan) and the GFP(1-10) sequence was subcloned from the split-GFP construct provided as a gift from Kristen Verhey (University of Michigan)[48]. The component was subcloned in pcDNA3.1 (−) as the vector backbone using primer sequences listed in Supplementary Table 1. GFP(1-10)-B14 was generated from cDNA isolated from HEK 293 T cells followed by PCR using primer sequences listed in Supplementary Table 1 and using GFP(1-10)-pcDNA3.1 (−) as the backbone. The remaining plasmids in this study used pcDNA3.1 (−) as the vector backbone and are generated from cDNA isolated from HEK 293 T cells followed by PCR using primer sequences listed in Supplementary Table 1. EMC4-FLAG was generated using specific primers whereas FLAG-EMC4 and FLAG-Δ50 EMC4 were kind gifts from Dr. Andrew Tai (University of Michigan). Protein tags (Myc, FLAG, or HA) located at either the N- or C-terminus are depicted as prefix or suffix, respectively. The siRNA-resistant EMC6-FLAG (EMC6∗-FLAG) and EMC7-FLAG (EMC7∗-FLAG) constructs were generated from EMC6-FLAG and EMC7-FLAG, respectively, by introducing silent mutations in the target site of EMC6 and EMC7 siRNA. Mutant EMC7-FLAG constructs were generated using siRNA-resistant EMC7-FLAG as a template and the specific primers listed in Supplementary Table 1. All plasmid constructs are verified by sequencing. Details of the recombinant DNA and oligonucleotides are listed in Supplementary Table 2.

**DNA transfection.** Fifty percent confluent CV-1 cells were transfected with the indicated plasmid using the FuGENE HD (Promega, Madison, WI) transfection reagent at a ratio of 1:4 (plasmid to transfection reagent; w/v) in the overexpression studies. For HEK 293 T cells, polyethylenimine (PEI; Polysciences, Warrington, PA) was used as the transfection reagent. Cells were transfected with the desired DNA construct for at least 24–48 h before the experiments were conducted.

**Preparation of viruses.** WT SV40 was prepared using OptiPrep gradient system. Briefly, SV40-infected or viral genome-transfected CV-1 cells were lysed in a buffer containing 50 mM Hepes (pH 7.5), 150 mM NaCl, and 0.5% Brij58 on ice for 30 min and centrifuged at $16,100 \times g$ for 10 min. The supernatant was loaded onto a discontinuous 20 and 40% OptiPrep gradient and centrifuged at 49,500 rpm for 2 h at 4 °C in an SW 55Ti rotor. A viral particle fraction between 20% and 40% OptiPrep was collected with a needle[17].

**Antibodies.** Mouse monoclonal SV40 large T-antigen antibody (working dilution 1:100 for IF), rabbit polyclonal Hsp90 (working dilution 1:2000 for WB) and Stx18 (working dilution 1:1000 for WB) antibodies were purchased from Santa Cruz Biotechnology (Santa Cruz, CA). Mouse monoclonal VP1 antibody (working dilution 1:2000 for WB and 1:500 for IF) was kindly provided by Walter Scott (University of Miami). Rabbit polyclonal BiP (working dilution 1:1000 for WB), EMC4 (working dilution 1:4000 for WB), SV40 VP2/3 (working dilution 1:500 for IF), Rab5 (working dilution 1:200 for WB), STARD3 (working dilution 1:20 for Immuno-EM) antibodies were purchased from Abcam (Cambridge, MA), whereas rabbit polyclonal anti-VP1 antibody (working dilution 1:2000 for WB and 1:500 for IF) was a gift from Harumi Kasamatsu (UCLA). Rat monoclonal BAP31 antibody (working dilution 1:3000 for WB and 1:250 for IF) was purchased from Thermo Fisher Scientific (Waltham, MA), and FLAG tag antibody (working dilution 1:5000 for WB, 1:500 for IF and 1:20 for Immuno-EM) was obtained from Sigma (St Louis, MO). Polyclonal CTA antibody (working dilution 1:2000 for WB) was produced against denatured CTA and generated by EMD Millipore (Burlington, MA). Mouse monoclonal Rab7 antibody (working dilution 1:500 for WB, 1:100 for IF and 1:20 for Immuno-EM) was purchased from Sigma (St. Louis, MO), whereas rabbit monoclonal Rab7 (working dilution 1:1000 for WB and 1:200 for IF) and EEA1 (working dilution 1:500 for IF) antibody were purchased from Cell Signaling Technology (Danvers, MA). Rabbit polyclonal EMC6 antibody (working dilution 1:2000 for WB) was purched from Aviva Systems Biology (San Diego, CA), while rabbit anti-EMC1 antibody (working dilution 1:2000 for WB) was purchased from Abgent (San Diego, CA). Antibody against protrudin (ZFYVE27) (working dilution 1:1000 for WB), monoclonal mouse GPF antibody (working dilution 1:10000 for WB), DnaJB14 antibody (working dilution 1:400 for IF) and polyclonal rabbit HA antibody (working dilution 1:5000 for WB and 1:500 for IF) were purchased from Proteintech Group (Chicago, IL). Mouse monoclonal anti-Myc antibody (working dilution 1:1000 for WB and 1:200 for IF) was a kind gift from Prof. Kristen Verhey (University of Michigan). F(ab') 2 Fragment of Goat-anti-Mouse IgG (H&L) (EM grade 15 nm), F(ab') 2 Fragment of Goat-anti-Rabbit IgG (H&L) (EM grade 6 nm), Goat-anti-Mouse IgG (H&L) (EM grade 6 nm), and Goat-anti-Rabbit IgG (H&L) (EM grade 15 nm) immunogold secondary antibodies (working dilution 1:25 for Immuno-EM) were purchased from Electron Microscopy Sciences (Hatfield, PA). Details of the antibodies are listed in Supplementary Table 2.

**Epifluorescence widefield microscopy, confocal microscopy, and structured illumination microscopy (SIM).** CV-1 cells were grown in 12-well plate followed by transfection with specific plasmids with FuGene (Promega) for 24 h where applicable. For knockdown studies, cells were reverse transfected with the desired siRNA using Lipofectamine RNAiMAX (Invitrogen) at the time of cell seeding. Cells washed with PBS followed by fixation with 4% formaldehyde at room temperature were then permeabilized using 0.2% Triton X-100, and blocked by 5% milk with 0.2% Tween. Primary antibodies were incubated for 1 h at room temperature or overnight at 4 °C (in the case of SIM), followed by incubation of fluorescent-conjugated secondary antibodies for 2 h at room temperature. Coverslips were mounted with ProLong Gold mounting medium (Thermo Fisher

Scientific) for epifluorescence widefield and confocal microscopy. In the case of SIM, coverslips were mounted with non-hardening Vectashield antifade mounting medium (Vector Laboratories, CA). Images were taken using inverted epifluorescence microscope (Nikon Eclipse TE2000-E) equipped with ×100 oil immersion objective (N.A. 1.4), Sola lumencore light engine and Photometrics CoolSnap HQ camera; Nikon A1 High Sensitivity (HS) Confocal microscope with CFI Apochromat TIRF ×60 oil immersion objective (N.A. 1.49), LU-N4/N4S 4-laser unit and two standard PMT, two GaAsP detectors; or Nikon N-SIM E in 3D-SIM mode with CFI SR HP Apochromat TIRF 100XC Oil immersion objective (NA 1.49), LU-NV series laser unit and ORCA-Flash 4.0 sCMOS camera (Hamamatsu Photonics K.K.). NIS-Elements C software was used to take confocal images and NIS-Elements AR software was used to take SIM images. For 561 channel, illumination modulation contrast was set to 1 and high-resolution noise suppression was set to 0.7 while for 488 channel, both illumination modulation contrast and high-resolution noise suppression was set to 1 during N-SIM stack reconstruction. FIJI distribution of ImageJ[49,50] was used for image processing, analyses, and assembly. FIJI Coloc2 plugin was used with Costes threshold regression to measure Mander's overlap coefficient and FIJI Colocalization threshold plugin was used to generate scatter plot and colocalization pixel map. SIMcheck FIJI/ImageJ plugin[51] was used to validate modulation contrast in SIM raw data and images with average modulation contrast to noise ratio higher than 6 were used for quantification. The analysis of reconstructed SIM images was done on a single z-plane.

**Knockdown-rescue experiments.** CV-1 cells were reverse transfected with specific siRNA using Lipofectamine RNAiMAX (Invitrogen). Twenty-four hour after siRNA transfection, cells were transfected with GFP-FLAG or WT or mutant EMC constructs. Twenty-four hour after DNA transfection, cells were infected with SV40 (MOI ~ 0.5) and at 20 hpi, cells were analyzed by epifluorescence widefield microscopy using anti SV40 T-antigen or anti VP2/VP3 and FLAG antibodies. To quantify infection, T-antigen-positive cells were scored in only those cells expressing the FLAG-tagged protein. To quantify VP2/VP3 exposure, mean VP2/VP3 signal intensity were measured by FIJI software (NIH) in only those cells expressing the FLAG-tagged protein.

**ER-to-cytosol membrane transport and ER-arrival assays.** These assays were performed as previously described[17]. Briefly, the indicated siRNA-treated CV-1 cells were infected with SV40 (MOI ~5) for 15 h. Cells were then incubated in HNp buffer (50 mM Hepes pH 7.5, 150 mM NaCl, and 1 mM PMSF) containing 0.1% digitonin at 4 °C for 10 min, and centrifuged at 20,000 × g for 10 min at 4 °C to generate a supernatant (cytosol) and a pellet (membrane) fraction. To isolate ER-localized SV40, the pellet fraction was further treated with HNp buffer containing 1% Triton X-100 for 10 min at 4 °C, and centrifuged at 20,000 × g for 10 min at 4 °C. The extracted supernatant material contains the ER-localized SV40. To assess ER-dependent formation of cholera toxin A1 (CTA1) subunit, CV-1 cells were treated with 10 nM CT (EMD Millipore, Darmstadt, Germany) for 90 min. Cells were harvested and lysed with HNp buffer containing 1% Triton X-100, 10 mM NEM for 10 min at 4 °C and immunoblotted under non-reducing condition.

**Immunoprecipitation.** Transfected cells were harvested using trypsin, and the cell pellets were washed three times with cold phosphate buffered saline (PBS, Gibco). Washed cells were lysed in HNp buffer (50 mM Hepes pH 7.5, 150 mM NaCl, and 1 mM PMSF) with 1% DBC (Deoxy Big CHAP) (Millipore) or 0.5% NP40 (Sigma) at 4 °C for 10 min. The cell extract was clarified by centrifugation at 20,000 × g for 10 min at 4 °C. The resulting extract was incubated overnight with a specific antibody against the endogenous protein and then incubated with protein A/G agarose beads. Alternatively, to pull down FLAG-tagged proteins, the cell extract was incubated with FLAG antibody-conjugated agarose beads (FLAG M2 beads) for 2 h at 4 °C. Samples were eluted with 1X SDS sample buffer with 1.25% β-mercaptoethanol (Sigma), and boiled for 10 min at 95 °C before subjected to SDS-PAGE and immunoblotting. Uncropped and unprocessed scans of blots are supplied in the Source Data file.

**Proximity-dependent biotin identification.** Fifty micromolar of biotin was added to HEK 293 T cells expressing either Myc-BioID2, Myc-BioID2-EMC4, or EMC7-BioID2-HA. After 16 h, cells were lysed with RIPA buffer (50 mM Tris pH 7.5, 150 mM NaCl, 1% SDS, 0.5% sodium deoxycholate, 1% Triton X100, and 1 mM PMSF) followed by affinity purification with Streptavidin C1 beads (Thermo Fisher Scientific) to capture biotinylated proteins. Next, the biotinylated proteins were eluted by biotin elution buffer (2X SDS sample buffer + 2 mM biotin) and boiled for 10 min at 95 °C before subjected to SDS-PAGE and immunoblotting with specific antibodies. Strdptavidin-HRP conjugate (Thermo Fisher Scientific) was used to assess the total biotinylated proteins.

**Quantification and statistical analysis.** All data obtained from at least three independent experiments (biological replicates) were combined for statistical analyses. Results were analyzed using Student two-tailed *t*-test. Data are represented as the mean values and error bar represents standard deviation (SD) ($n \geq 3$) where indicated. $p < 0.05$ was considered to be significant.

**Reporting summary.** Further information on research design is available in the Nature Research Reporting Summary linked to this article.

## Data availability
Data that support the findings of this study are available from the corresponding author upon reasonable request. The source data underlying Fig. 1; 2A-B, F, G; 3B; 4; 5B; 6D; 7B-E; 8A-G and Supplementary Figures 1B; 3; 5C; and 7B are provided as a Source Data file.

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

## Acknowledgements

B.T. is supported by the National Institutes of Health (RO1AI064296 and R21AI140449). B.T. and L.Q. are funded by National Institute of Health (RO1DK111174). We also want to thank Aaron Taylor (University of Michigan), Woo Jung Cho (University of Michigan) and Sasha Meshinci (University of Michigan) for their help in SIM and confocal and electron microscopy.

## Author contributions

P.B.: Conceptualization, data curation, formal analysis, investigation, methodology, writing—original draft, writing—review and editing; B.T.: Conceptualization, data curation, formal analysis, supervision, funding acquisition, investigation, writing—original draft, writing—review and editing. M.T. and L.Q. assisted in the electron microscopy (EM) experiments.

## Competing interests

The authors declare no competing interests.
