## [Peer Review File · Nature Communications]

Reviewers' comments:

Reviewer #1 (Remarks to the Author):

SV40 has been an outstanding model to study normal cell biological processes for decades. In this manuscript Bagchi and Tsai discover an important cellular mechanism that allows late endosomal cargo, in their case SV40 virus, to access the endoplasmic reticulum during entry. This is a critical step in the infectious process as SV40 requires the environment of the ER to begin the uncoating process that eventually leads to successful infection. Many viruses and cargos that require such a step most likely utilize this novel mechanism. This mechanism was uncovered by an exhaustive and well controlled series of experiments that not only identified the principle players but also identified the mechanism. The first set of experiments involved knocking down ER-membrane complex proteins (EMC) 4, 6, and 7 using siRNA. Knockdown of each led to reduced infection by SV40 that could be rescued by transfecting cells with an siRNA resistant construct of EMC4, 6, and 7. Appropriate controls were used in all cases. They next used an assay that relies on selectively permeabilizing the plasma membrane while leaving internal membranes intact. They found that knockdown of all three resulted in less cytosolic VP1. This is not input VP1 but rather VP1 that had presumably trafficked to the ER and then was extruded into the cytosol as a partially uncoated virus particle. To be sure that this was the case they measured ER associated VP1 and found that EMC4 and 7 resulted in less VP1 getting to the ER. This was not the case for EMC6 and they speculate that this is likely involved in ER exit to the cytosol. To confirm these data they imaged exposure of the internal viral proteins VP2/3 as this only occurs once the virus arrives in the ER. They found that EMC6 depleted cells scored positive for VP2/3 exposure but that EMC4 and 7 depleted cells did not confirming the fact that the virus never reached the ER. They then used BioID to prove that EMC4 and 7 were 1. Localized to the ER membrane and 2. Interacted with late endosome associated Rab7. This was also shown using structured illumination microscopy. The domains responsible for this tethering were localized to a low complexity Rab 7 binding domain present in the first 50 AA of EMC 4 and in the C-terminus of EMC7. How does this all work? EMC4 and 7 also bind the snare proteins, STX18, to promote small vesicle fusion between the LE and the ER.

In summary, this is an outstanding, novel, and important contribution to not only the SV40 and virology communities but also to the broader cell biological community as well. The experiments are well controlled and the conclusions are supported by their data.

Reviewer #2 (Remarks to the Author):

In their manuscript Bagchi & Tsai combine biochemical and high-resolution imaging approaches to investigate polyomavirus SV40 transfer from late endosomes (LE) to the endoplasmic reticulum (ER), a critical step in the SV40 infection pathway. Using an RNAi knockdown strategy, they identify

the ER membrane complex (EMC) proteins EMC 4 and EMC 7, but not EMC 6, as key factors in this process mediating the tethering of LE via Rab7 to ER also involving the ER membrane protein syntaxin 18.

Overall the manuscript is clearly and concisely written, experiments are well conceived and the main conclusions are largely supported by the presented data. As an outsider to the field, I refrain from commenting on the novelty or relevance of the biological findings. My comments below therefore mainly refer to the technical aspects of the high-resolution imaging employed, which exhibit some weaknesses as outlined below.

Major issues:

The authors use hardening mounting media ProLong Gold or ProLong Glass. With progressing curing of those media, the embedded cells get increasingly compressed, which can distort cellular features and may affect the 3D spatial relationships of target structures. Hence it would be advisable that the authors provide suitable control experiments that demonstrate similar degrees of colocalisation using a 3D structure preserving glycerol based mounting medium (e.g. Vectashield, Slowfade).

Generally, the descriptions of the different imaging methods lack important details, e.g. on hardware (objectives, laser lines) and software settings (SIM reconstruction). Furthermore, the authors need to specify which Fiji/ImageJ functions or plugins and corresponding settings they have been using for their quantitative analyses. E.g., the Mander's overlap coefficient or the number of overlapping pixels, respectively, is highly dependent on the threshold settings.

It is unclear whether and how chromatic shift/dispersion were corrected for, as this is crucial for the validity of colocalisation experiments. Furthermore, the authors need to specify whether analyses were performed in 3D, on a single z-plane or whether they used maximum intensity projections. They should also indicate this for the images displayed.

Fig. 5 presents a key evidence based on super-resolution 3D-SIM data. However, I am not convinced regarding the data quality, and correspondingly the quantification shown in panel B. In particular dense cellular features such as the ER can be affected by physical compression by the mounting condition (see above). Such structures are also particular prone to SIM specific reconstruction artifacts (see Demmerle et al. 2017, Nat Protocols), such as high-frequency noise artefacts that may have been compensated by (over)filtering the data at the expense of the effectively achieved resolution.

Thus, I strongly recommend to use the SIMcheck ImageJ plugin (Ball et al. 2015, Sci Reports), in particular to validate the modulation contrast in the raw data (responsible for gaining high-frequency information) and the effective resolution increase in the reconstructed data.

I further recommend to show profile plots of colocalisation events of the (unsaturated) reconstructed images to have a complementary quantitative readout.

Minor issues:

Line 70 ff. “EMC4 and EMC7 accomplish this by engaging the cytosolic Rab7 GTPase bound to the LE membrane, presumably to support membrane contact between the LE and ER. EMC4”. It would be interesting to know if LE-ER membrane contact is also occurring in non-infected cells, or if these contacts are specifically triggered by the virus infection.

Line 74: “...suggest that EMC4 and EMC7 function as novel molecular tethers...”. My understanding is, that the function itself is not novel but the discovery of the function.

Line 75 “...connecting two different intracellular organelles to enable efficient transport of a viral particle between these compartments required for infection.” Does the virus not rather harness this function for its benefit?

Line 492: Here as well as in some other places the authors mistakenly use the term “immunofluorescence microscopy”. Instead it should read “widefield microscopy” or “epifluorescence widefield microscopy”. “Immunofluorescence” refers to the sample preparation, not the imaging modality.

Line 496: “...by fixation with 4% paraformaldehyde” should read “by fixation with 4% formaldehyde” as paraformaldehyde is the polymerisation product of formaldehyde.

Line 506: Here and in the figure legends the authors refer to have used “Fiji software (NIH)” or “Fiji (NIH)”. Instead they should initially refer to the “Fiji distribution of ImageJ (Schindelin, 2012, Nat Meth; Schneider et al. 2012, Nat Meth) was used...” and later simply to “Fiji/ImageJ”.

Fig. 2E: Having the DAPI signal displayed in all panels is redundant and distracts from the main message, the absence of green VP2/VP3 signal in EMC4 and EMC7 knockdowns. V1 signal appears oversaturated. Generally, oversaturation should be avoided unless it is used to highlight specific features (in this case it needs to be indicated in the legend). As the panel is read from top left to bottom right, I would recommend to move the annotations accordingly to the top and left side of the panel.

Reviewer #3 (Remarks to the Author):

The authors show that the EMC subunits EMC4 and EMC7 are critical for SV40 entry and replication. Moreover, they demonstrate very well that ER-resident EMC4 and EMC7 interact with the late endosomal Rab7 in its GTP-bound state. As they claim EMC4 & EMC7 form membrane contact sites between late endosomes and the ER that facilitate viral particle transfer, their results fail to prove this notion. Following are the points of concern:

1- Authors fail to show that EMC4 and EMC7 form membrane contact sites (MCS) between the ER and late endosomes (LE). For this, they must show by electron microscopy that EMC4 and EMC7 localize at MCS between the ER and SV40-containing LE. Also, if EMC proteins form MCS between the ER and LE by interacting with Rab7 [QL], overexpression of Rab7[QL] with EMC proteins must increase the MCS between LE and ER.

2- How sure are the authors that EMC proteins facilitate transfer of virus particles directly from LE to the ER? Authors also mention that “upon endocytosis, SV40 reaches early endosome and then LE, from where it bypasses the Golgi en route to the ER”. In their experiment they do not rule out the trafficking via Golgi apparatus.

3- In Figure 2C, authors claim to detect the ER-bound VP1 by solubilizing membrane fractions in 1% TritonX-100. However, cells contain various organelle membranes that are soluble in 1% TritonX-100, e.g. Golgi stacks. Therefore, the only conclusion they can draw is that EMC4 and EMC7 are required for VP1 to be trafficked to Triton-soluble organelles. To show that their solubilisation method is specific for the ER membranes, they must show that the ER marker is soluble in 1% TritonX-100 and other organelle markers -such as early endosome, late endosome, ERGIC, Golgi and the plasma membrane- are not present in the detergent extracts.

4- In Figure 2E, siEMC4 and siEMC7 cells have a reduced number of VP1-positive structures. Could it be that these cells have less internalization of the virus particles?

5- In Figure 3A, siEMC4 and siEMC7 cells contain more Rab7-positive compartments. Could it be that increased co-localization is an artefact of this? Also, for figure 3B, authors should use a pixel-based colocalization such as Pearson's correlation coefficient.

6- SIM images are simply not convincing enough to claim that siEMC4+siEMC7 cells have less MCS.

7- The mechanism of how interaction with Rab7 facilitates SV40 infection is missing.

Also some minor issues are:

1- Authors should explain what BK PyV JC PyV stand for?

2- In Figure 3A, authors must not change the aspect ratio of white boxes and zoom-in boxes.

3- Authors say: “the EMC4, EMC6, and EMC7 siRNAs used in our experiments specifically decreased the expression of the intended EMC subunit while largely maintaining the stability of the other subunits.”. It seems siEMC4 increases EMC7 levels in Figure 1D; siEMC6 decreases EMC4 levels in Figure 1E; siEMC7 decreases EMC4 levels in Figure 1D and 1E, decreases EMC6 levels in Figure 1D, and increases EMC6 levels in Figure 1E. The experiments must be repeated and quantified to draw more solid conclusions.

Reviewer #1 (Remarks to the Author):

SV40 has been an outstanding model to study normal cell biological processes for decades. In this manuscript Bagchi and Tsai discover an important cellular mechanism that allows late endosomal cargo, in their case SV40 virus, to access the endoplasmic reticulum during entry. This is a critical step in the infectious process as SV40 requires the environment of the ER to begin the uncoating process that eventually leads to successful infection. Many viruses and cargos that require such a step most likely utilize this novel mechanism. This mechanism was uncovered by an exhaustive and well controlled series of experiments that not only identified the principle players but also identified the mechanism. The first set of experiments involved knocking down ER-membrane complex proteins (EMC) 4, 6, and 7 using siRNA. Knockdown of each led to reduced infection by SV40 that could be rescued by transfecting cells with an siRNA resistant construct of EMC4, 6, and 7. Appropriate controls were used in all cases. They next used an assay that relies on selectively permeabilizing the plasma membrane while leaving internal membranes intact. They found that knockdown of all three resulted in less cytosolic VP1. This is not input VP1 but rather VP1 that had presumably trafficked to the ER and then was extruded into the cytosol as a partially uncoated virus particle. To be sure that this was the case they measured ER associated VP1 and found that EMC4 and 7 resulted in less VP1 getting to the ER. This was not the case for EMC6 and they speculate that this is likely involved in ER exit to the cytosol. To confirm these data they imaged exposure of the internal viral proteins VP2/3 as this only occurs once the virus arrives in the ER. They found that EMC6 depleted cells scored positive for VP2/3 exposure but that EMC4 and 7 depleted cells did not confirming the fact that the virus never reached the ER. They then used BiOID to prove that EMC4 and 7 were 1. Localized to the ER membrane and 2. Interacted with late endosome associated Rab7. This was also shown using structured illumination microscopy. The domains responsible for this tethering were localized to a low complexity Rab 7 binding domain present in the first 50 AA of EMC 4 and in the C-terminus of EMC7. How does this all work? EMC4 and 7 also bind the snare proteins, STX18, to promote small vesicle fusion between the LE and the ER.

In summary, this is an outstanding, novel, and important contribution to not only the SV40 and virology communities but also to the broader cell biological community as well. The experiments are well controlled and the conclusions are supported by their data.

We thank this reviewer for the positive comments. Because NO experiments and questions were raised, we did not specifically address this reviewer's concern.

Reviewer #2 (Remarks to the Author):

In their manuscript Bagchi & Tsai combine biochemical and high-resolution imaging approaches to investigate polyomavirus SV40 transfer from late endosomes (LE) to the endoplasmic reticulum (ER), a critical step in the SV40 infection pathway. Using an RNAi knockdown strategy, they identify the ER membrane complex (EMC) proteins

EMC 4 and EMC 7, but not EMC 6, as key factors in this process mediating the tethering of LE via Rab7 to ER also involving the ER membrane protein syntaxin 18.

Overall the manuscript is clearly and concisely written, experiments are well conceived and the main conclusions are largely supported by the presented data. As an outsider to the field, I refrain from commenting on the novelty or relevance of the biological findings. My comments below therefore mainly refer to the technical aspects of the high-resolution imaging employed, which exhibit some weaknesses as outlined below.

Major issues:

1. The authors use hardening mounting media ProLong Gold or ProLong Glass. With progressing curing of those media, the embedded cells get increasingly compressed, which can distort cellular features and may affect the 3D spatial relationships of target structures. Hence it would be advisable that the authors provide suitable control experiments that demonstrate similar degrees of colocalisation using a 3D structure preserving glycerol based mounting medium (e.g. Vectashield, Slowfade).

We thank the reviewer for raising this point. In the revised manuscript, we repeated the same SIM experiment using the glycerol-based mounting medium Vectashield, as requested. Additionally, we used (transfected) STARD3, a late endosome (LE) transmembrane protein, as a LE marker (instead of Rab7) because it showed a more distinct LE morphology. Importantly, under these new conditions, we still found that depletion of EMC4 and EMC7 reduced the level of co-localization between ER and LE (Figure 5A); when quantified by the Mander's Overlap Coefficient, the extent of co-localization was reduced by more than a half fold under the knockdown condition (Figure 5B), a result similar to what we initially reported in the original manuscript. This finding further supports the idea that these EMC subunits support co-localization of the ER and LE.

2. Generally, the descriptions of the different imaging methods lack important details, e.g. on hardware (objectives, laser lines) and software settings (SIM reconstruction). Furthermore, the authors need to specify which Fiji/ImageJ functions or plugins and corresponding settings they have been using for their quantitative analyses. E.g., the Mander's overlap coefficient or the number of overlapping pixels, respectively, is highly dependent on the threshold settings.

As requested, we now include all of these information (lines 569-600) in the method section of the revised manuscript.

3. It is unclear whether and how chromatic shift/dispersion were corrected for, as this is crucial for the validity of colocalisation experiments. Furthermore, the authors need to specify whether analyses were performed in 3D, on a single z-plane or whether they used maximum intensity projections. They should also indicate this for the images displayed.

We did not correct for the chromatic shift. However, the objective used for the SIM experiments is CFI SR HP Achromat TIRF 100XC Oil (Nikon), and according to the manufacture, this objective is “aligned and inspected using wave front aberration measurement technologies to ensure the lowest possible asymmetric aberration with superb optical performance required for super-resolution imaging”.

In the revised manuscript, we specified that the analyses were performed on a single z-plane (line 599-600). We also indicated this information in the figure legends (line 707-708, 718-719, 748, 759-760, 858, 898-899).

4. Fig. 5 presents a key evidence based on super-resolution 3D-SIM data. However, I am not convinced regarding the data quality, and correspondingly the quantification shown in panel B. In particular dense cellular features such as the ER can be affected by physical compression by the mounting condition (see above). Such structures are also particular prone to SIM specific reconstruction artifacts (see Demmerle et al. 2017, Nat Protocols), such as high-frequency noise artefacts that may have been compensated by (over)filtering the data at the expense of the effectively achieved resolution.

As requested, we used the glycerol-based mounting reagent Vectashield in the revised manuscript (see point #1 above), and provided higher quality data (Figure 5).

5. Thus, I strongly recommend to use the SIMcheck ImageJ plugin (Ball et al. 2015, Sci Reports), in particular to validate the modulation contrast in the raw data (responsible for gaining high-frequency information) and the effective resolution increase in the reconstructed data. I further recommend to show profile plots of colocalisation events of the (unsaturated) reconstructed images to have a complementary quantitative readout.

As requested, in this revised manuscript, SIMcheck FIJI/ImageJ plugin was used to validate the modulation contrast in the SIM raw data. Specifically, only images with average modulation contrast to noise ratio higher than 6 were used for quantification analysis (line 597-599). Furthermore, as requested, we now show the scatter plots and the co-localization pixel maps of reconstructed images corresponding to the scrambled and EMC4+EMC7 siRNA-treated samples in the revised manuscript as complementary quantitative readout (Supplemental Figure 4).

Minor issues:

1. Line 70 ff. “EMC4 and EMC7 accomplish this by engaging the cytosolic Rab7 GTPase bound to the LE membrane, presumably to support membrane contact between the LE and ER. EMC4”. It would be interesting to know if LE-ER membrane contact is also occurring in non-infected cells, or if these contacts are specifically triggered by the virus infection.

Our BioID and co-immunoprecipitation experiments were performed in the absence of SV40 infection (Figure 4). Thus, the EMC4- and EMC7-mediated LE-ER contact is

present even without SV40 infection. We believe SV40 is simply taking advantage of this pre-existing LE-ER contact during entry.

2. Line 74: "...suggest that EMC4 and EMC7 function as novel molecular tethers...". My understanding is, that the function itself is not novel but the discovery of the function.

In this manuscript, we identified two new molecular tethers (i.e. EMC4 and EMC7) between the ER and LE which were not previously known. Nonetheless, to avoid confusion, we changed the term 'function' to 'act' in the revised manuscript (line 76).

3. Line 75 "...connecting two different intracellular organelles to enable efficient transport of a viral particle between these compartments required for infection." Does the virus not rather harness this function for its benefit?

Yes, we do believe that SV40 is simply harnessing this pre-existing function for its benefit during infection (i.e. transport from the LE to the ER).

4. Line 492: Here as well as in some other places the authors mistakenly use the term "immunofluorescence microscopy". Instead it should read "widefield microscopy" or "epifluorescence widefield microscopy". "Immunofluorescence" refers to the sample preparation, not the imaging modality.

As requested, we have changed the text in the revised manuscript accordingly (line 569, 580, 606, 684, 774, 779, 784, 866, 877-878, 920).

5. Line 496: "...by fixation with 4% paraformaldehyde" should read "by fixation with 4% formaldehyde" as paraformaldehyde is the polymerisation product of formaldehyde.

In the revised manuscript, the text is modified as suggested (line 575).

6. Line 506: Here and in the figure legends the authors refer to have used "FIJI software (NIH)" or "FIJI (NIH)". Instead they should initially refer to the "Fiji distribution of ImageJ (Schindelin, 2012, Nat Meth; Schneider et al. 2012, Nat Meth) was used..." and later simply to "Fiji/ImageJ".

In the revised manuscript, the text is modified as suggested (line 594 and in the modified figure legends).

7. Fig. 2E: Having the DAPI signal displayed in all panels is redundant and distracts from the main message, the absence of green VP2/VP3 signal in EMC4 and EMC7 knockdowns. V1 signal appears oversaturated. Generally, oversaturation should be avoided unless it is used to highlight specific features (in this case it needs to be indicated in the legend). As the panel is read from top left to bottom right, I would recommend to move the annotations accordingly to the top and left side of the panel.

In the revised manuscript, we used lower exposure image of VP1 to avoid oversaturation and changed the image corresponding to the scrambled siRNA treatment. The presentation of Figure 2E is also modified as suggested.

Reviewer #3 (Remarks to the Author):

The authors show that the EMC subunits EMC4 and EMC7 are critical for SV40 entry and replication. Moreover, they demonstrate very well that ER-resident EMC4 and EMC7 interact with the late endosomal Rab7 in its GTP-bound state. As they claim EMC4 & EMC7 form membrane contact sites between late endosomes and the ER that facilitate viral particle transfer, their results fail to prove this notion. Following are the points of concern:

1- Authors fail to show that EMC4 and EMC7 form membrane contact sites (MCS) between the ER and late endosomes (LE). For this, they must show by electron microscopy that EMC4 and EMC7 localize at MCS between the ER and SV40-containing LE. Also, if EMC proteins form MCS between the ER and LE by interacting with Rab7 [QL], overexpression of Rab7[QL] with EMC proteins must increase the MCS between LE and ER.

We have performed several additional experiments to address this valid concern:

First, we introduced a new split-GFP assay to assess the effect of EMC4 and EMC7 knockdown in LE-ER contact (Figure 6 and Supplemental Figure 5). Our new results demonstrate that depletion of EMC4 and EMC7 reduced the GFP signal, suggesting that these two EMC subunits promote LE-ER contact.

Second, to further strengthen this point, we used immuno-EM analysis (as requested) and found that EMC4 and EMC7 are present at (or close to) the LE-ER contact site (Supplemental Figure 6), in agreement with the idea that these membrane proteins support LE-ER contact.

Third, we repeated the super-resolution microscopy (SIM) experiments - as suggested by reviewer #2 - to obtain higher quality data. Our new results clearly showed that depletion of EMC4 and EMC7 decreased the level of ER-LE co-localization (Figure 5 and Supplemental Figure 4). Again, this is consistent with the split-GFP and immuno-EM analysis, strongly suggesting that these EMC proteins support ER-LE contact.

We hope that these three new findings, in conjunction with the BioID results revealing that EMC4 and EMC7 are in close proximity (10 nm range) to the LE-associated Rab7 (Figure 4A) and the co-IP data demonstrating that EMC4 and EMC7 bind to the membrane-bound form of Rab7 (Figure 4C and 4E), make a compelling case that these EMC subunits indeed act as molecular tethers to support contact between the ER and LE.

Because other ER membrane proteins (such as protrudin) also interact with LE-associated Rab7 to establish LE-ER membrane contact (Raiborg C. et al, Nature, 2015), we do not believe that overexpression of Rab7(QL) can conclusively demonstrate that the EMC proteins form tethers between ER and LE.

2- How sure are the authors that EMC proteins facilitate transfer of virus particles directly from LE to the ER? Authors also mention that “upon endocytosis, SV40 reaches early endosome and then LE, from where it bypasses the Golgi en route to the ER”. In their experiment they do not rule out the trafficking via Golgi apparatus.

It is widely established that SV40 and other polyomaviruses reach the ER from the LE and do not traffic to the ER via the Golgi apparatus. This phenomenon has been previously reported by many groups, including ours (Norkin et. al., Virol. J., 2005; Gilbert and Benjamin, J. Virol., 2004; Qian et. al., Plos Pathogens, 2009; Kartenbeck et. al., J Cell. Biol., 1989; Zhao et. al., mSphere, 2017).

3- In Figure 2C, authors claim to detect the ER-bound VP1 by solubilizing membrane fractions in 1% TritonX-100. However, cells contain various organelle membranes that are soluble in 1% TritonX-100, e.g. Golgi stacks. Therefore, the only conclusion they can draw is that EMC4 and EMC7 are required for VP1 to be trafficked to Triton-soluble organelles. To show that their solubilisation method is specific for the ER membranes, they must show that the ER marker is soluble in 1% Triton X-100 and other organelle markers -such as early endosome, late endosome, ERGIC, Golgi and the plasma membrane- are not present in the detergent extracts.

Our lab previously developed this cell-based, semi-permeabilization assay to isolate cytosol and ER-localized SV40 (Inoue and Tsai, Plos Pathogens, 2011). During entry, SV40 binds to ganglioside GM1 at the cell surface and traffics with this lipid receptor all the way to the ER. Only upon reaching the ER is SV40 released into the ER lumen. Ganglioside GM1 is highly enriched in lipid rafts, and factors (including viruses) bound to GM1 also partition into lipid rafts. Because proteins in lipid rafts are not extracted by Triton X-100, SV40 bound to GM1 is therefore unable to be extracted by this detergent. However, when SV40 is released from GM1 – which only occurs when the virus reaches the ER – the virus becomes detergent extractable. This is the basis of this assay. Hence, the ER-localized VP1 in Figure 2C refers only to the presence/absence of SV40, but not any membrane markers, in this fraction.

We have used this assay to demonstrate ER-localized SV40 previously (e.g. Bagchi et al., eLife, 2016; Ravindran et al., Nature Communications, 2017; Inoue and Tsai, PloS Pathogen, 2017).

4- In Figure 2E, siEMC4 and siEMC7 cells have a reduced number of VP1-positive structures. Could it be that these cells have less internalization of the virus particles?

We do not think this is the case. Instead, as pointed out by reviewer #2, this could be a case of an oversaturation issue. We have now fixed the oversaturation problem by

using a less exposed version of VP1 in the revised manuscript – the new images show almost similar amount of VP1 signal in all conditions (Figure 2E in revised manuscript). It is important to note that the VP2/3 quantification data in Figure 2F was calculated by normalizing against the VP1 signal, thereby ensuring that no VP1 signal difference can affect the conclusion.

Figure 3A also demonstrates no change in the VP1 signal in siEMC4 and siEMC7 treated cells when compared to the scrambled siRNA or siEMC6 treated cells. Furthermore, our cell-based semi-permeabilization assay findings (Figure 2A and 2C) show a similar amount of VP1 in the membrane fractions of siEMC4 and siEMC7 treated cells when compared to the other two conditions.

5- In Figure 3A, siEMC4 and siEMC7 cells contain more Rab7-positive compartments. Could it be that increased co-localization is an artefact of this? Also, for figure 3B, authors should use a pixel-based colocalization such as Pearson's correlation coefficient.

In the revised manuscript, we provide new images of the siEMC4 and siEMC7 knockdown conditions that show similar Rab7 positive signals when compared to the control and siEMC6 knockdown conditions (Figure 3A). Additionally, as requested, we used pixel-based Mander's Overlap Coefficient for quantification (Figure 3B), which demonstrates a similar result as the original manuscript.

6- SIM images are simply not convincing enough to claim that siEMC4+siEMC7 cells have less MCS.

This is a concern expressed by reviewer #2 (major point #1).

Again, in the revised manuscript, we repeated the SIM experiment using the glycerol-based mounting medium Vectashield (as requested by reviewer #2). We also used (transfected) STARD3, a LE transmembrane protein, as a LE marker (instead of Rab7) since it showed a more distinct LE morphology. Importantly, under these new conditions, we still found that depletion of EMC4 and EMC7 reduced the level of co-localization between ER and LE (Figure 5A); when quantified by the Mander's Overlap Coefficient, the extent of co-localization was reduced by more than a half fold under the knockdown condition (Figure 5B), a result similar to what we reported in the original manuscript. This finding further supports the idea that these EMC subunits support co-localization of the ER and LE.

Additionally, as part of the SIM analysis, we show the scatter plots and the co-localization pixel maps of reconstructed images corresponding to the scrambled and EMC4+EMC7 siRNA-treated samples in the revised manuscript as complementary quantitative readout (Supplemental Figure 4).

7- The mechanism of how interaction with Rab7 facilitates SV40 infection is missing.

Our data showing that mutant EMC4 and EMC7 which cannot interact with LE-associated Rab7 are defective in SV40 infection and VP2/3 exposure (Figure 7A-7D) suggest that EMC4 and EMC7 interaction with Rab7 is required for SV40 ER-arrival from the cell surface, a decisive SV40 infection step. Our model posits that by binding to LE-associated Rab7, EMC4 and EMC7 tether the ER to the LE – this presumably allows efficient delivery of LE-localized SV40 into the ER.

Nonetheless, to provide more mechanistic insight, we performed an additional experiment to strengthen the importance of LE-associated Rab7 during SV40 infection in the revised manuscript. Specifically, our new data shows that when compared to cells expressing WT Rab7, cells expressing dominant-negative Rab7 (that cannot associate with the LE membrane) markedly blocked ER-arrival of SV40 from the cell surface (Figure 7E). This finding demonstrates that LE-associated Rab7 is crucial for SV40 to arrive in the ER. And because EMC4 and EMC7 can only interact with membrane-associated Rab7, this result further underscores the importance of the EMC4/EMC7-Rab7 interaction during SV40 entry.

Also some minor issues are:

1- Authors should explain what BK PyV JC PyV stand for?

BK and JC stand for the initials of the individuals from which the respective virus was isolated. Due to privacy concerns, this information should not be included in the manuscript.

2- In Figure 3A, authors must not chance the aspect ratio of white boxes and zoom-in boxes.

As requested, the image is modified in the revised manuscript (Figure 3A).

3- Authors say: “the EMC4, EMC6, and EMC7 siRNAs used in our experiments specifically decreased the expression of the intended EMC subunit while largely maintaining the stability of the other subunits.”. It seems siEMC4 increases EMC7 levels in Figure 1D; siEMC6 decreases EMC4 levels in Figure 1E; siEMC7 decreases EMC4 levels in Figure 1D and 1E, decreases EMC6 levels in Figure 1D, and increases EMC6 levels in Figure 1E. The experiments must be repeated and quantified to draw more solid conclusions.

As requested, the experiments were repeated three times and the revised manuscript includes the quantifications (Figure 1D and 1E, bottom panels).

REVIEWERS' COMMENTS:

Reviewer #2 (Remarks to the Author):

The authors have satisfactorily addressed most of my concerns.

In the revised manuscript I only noted the the following minor issues:

1) In the all figure legends referring to quantitative co-localisation analyses, the authors state that the "values represent the mean \pm SD of three independent experiments". It may be useful to also state (al least roughly) how many cells where analysed for each experiment.

2) In line 594/597 the references 49 and 50 are interchanged.

Reviewer #3 (Remarks to the Author):

The authors addressed the points. But there is one point that the authors should consider. The direct IP of Rab7 through EMC4 (Fig 4B) is extraordinary minor. It could be that the interaction is not direct with Rab7 but indirect through one of the Rab7 effectors or complexes such as HOPS. The authors should include this in their model and discussion. Even the GFP complementation experiments do not exclude such a model and co-incidental interactions can be stabilized by this fluorescent trick. The EM also does not show that Rab7 colocalizes with the EMC proteins (in fact, they are separated in space). Hence this requires a more subtle discussion.

REVIEWERS' COMMENTS:

Reviewer #2 (Remarks to the Author):

The authors have satisfactorily addressed most of my concerns.

In the revised manuscript I only noted the the following minor issues:

1) In the all figure legends referring to quantitative co-localisation analyses, the authors state that the "values represent the mean \pm SD of three independent experiments". It may be useful to also state (at least roughly) how many cells were analysed for each experiment.

Authors' Response: The approximate number of cells analyzed for each co-localization experiments are now included in the respective figure legends of the revised manuscript.

2) In line 594/597 the references 49 and 50 are interchanged.

Authors' Response: Thank you. This has been changed in the revised manuscript.

Reviewer #3 (Remarks to the Author):

The authors addressed the points. But there is one point that the authors should consider. The direct IP of Rab7 through EMC4 (Fig 4B) is extraordinary minor. It could be that the interaction is not direct with Rab7 but indirect through one of the Rab7 effectors or complexes such as HOPS. The authors should include this in their model and discussion. Even the GFP complementation experiments do not exclude such a model and co-incidental interactions can be stabilized by this fluorescent trick. The EM also does not show that Rab7 colocalizes with the EMC proteins (in fact, they are separated in space). Hence this requires a more subtle discussion.

Authors' Response: The Discussion section of the revised manuscript has been modified as requested. In addition, we also modified the model accordingly.